# NetKet 3: Machine Learning Toolbox for Many-Body Quantum Systems

Filippo Vicentini[1]⋆, Damian Hofmann[2], Attila Szabó[3,4], Dian Wu[1], Christopher Roth[5], Clemens Giuliani[1], Gabriel Pescia[1], Jannes Nys[1], Vladimir Vargas-Calderón[6], Nikita Astrakhantsev[7] and Giuseppe Carleo[1]

**1** École Polytechnique Fédérale de Lausanne (EPFL), Institute of Physics,
CH-1015 Lausanne, Switzerland
**2** Max Planck Institute for the Structure and Dynamics of Matter,
Center for Free-Electron Laser Science (CFEL),
Luruper Chaussee 149, 22761 Hamburg, Germany
**3** Rudolf Peierls Centre for Theoretical Physics,
University of Oxford, Oxford OX1 3PU, United Kingdom
**4** ISIS Facility, Rutherford Appleton Laboratory,
Harwell Campus, Didcot OX11 0QX, United Kingdom
**5** Physics Department, University of Texas at Austin
**6** Grupo de Superconductividad y Nanotecnología, Departamento de Física,
Universidad Nacional de Colombia, Bogotá, Colombia
**7** Department of Physics, University of Zurich, CH-8057 Zurich, Switzerland
⋆ filippo.vicentini@epfl.ch

May 27, 2022

## Abstract

**We introduce version 3 of NetKet, the machine learning toolbox for many-body quantum physics. NetKet is built around neural quantum states and provides efficient algorithms for their evaluation and optimization. This new version is built on top of JAX, a differentiable programming and accelerated linear algebra framework for the Python programming language. The most significant new feature is the possibility to define arbitrary neural network ansätze in pure Python code using the concise notation of machine-learning frameworks, which allows for just-in-time compilation as well as the implicit generation of gradients thanks to automatic differentiation. NetKet 3 also comes with support for GPU and TPU accelerators, advanced support for discrete symmetry groups, chunking to scale up to thousands of degrees of freedom, drivers for quantum dynamics applications, and improved modularity, allowing users to use only parts of the toolbox as a foundation for their own code.**

# 1 Introduction

During the last two decades, we have witnessed tremendous advances in machine learning (ML) algorithms which have been used to solve previously difficult problems such as image recognition [1, 2] or natural language processing [3]. This has only been possible thanks to sustained hardware development: the last decade alone has seen a 50-fold increase in available computing power [4]. However, unlocking the full computational potential of modern arithmetic accelerators, such as GPUs, used to require significant technical skills, hampering researchers in their efforts. The incredible pace of algorithmic advances must therefore be attributed, at least in part, to the development of frameworks allowing researchers to tap into the full potential of computer clusters while writing high-level code [5, 6].

In the last few years, researchers in quantum physics have increasingly utilized machine-learning techniques to develop novel algorithms or improve on existing approaches [7]. In the context of variational methods for many-body quantum physics in particular, the method of *neural quantum states* (NQS) has been developed [8]. NQS are based on the idea of using neural networks as an efficient parametrization of the quantum wave function. They are of particular interest because of their potential to represent highly entangled states in more than one dimension with polynomial resources [9], which is a significant challenge for more established families of variational states. NQS are also flexible: they have been successfully used to determine variational ground states of classical [10] and quantum Hamiltonians [11–18] as well as excited states [13], to approximate Hamiltonian unitary dynamics [8, 19–24], and to solve the Lindblad master equation [25–27]. In particular, NQS are currently used in the study of frustrated quantum systems [13, 16–18, 28–31], which have so far been challenging to optimize by established numerical techniques. They have also been used to perform tomographic state reconstruction [32] and efficiently approximate quantum circuits [33].

A complication often encountered when working with NQS is, however, that standard ML frameworks like Tensorflow [34] or PyTorch [35] are not geared towards these kind of *quantum mechanical* problems, and it often takes considerable technical expertise to use them for such non-standard tasks. Alternatively, researchers sometimes avoid those frameworks and implement their routines from scratch, but this often leads to sub-optimal performance. We believe that it is possible to foster research at this intersection of quantum physics and ML by providing an easy-to-use interface exposing quantum mechanical objects to ML frameworks.

We therefore introduce version 3 of the NETKET framework [36].[1] NETKET 3 is an open-source Python toolbox expressing several quantum mechanical primitives in the differentiable programming framework JAX [5, 37]. NETKET provides an easy-to-use interface to high-performance variational techniques without the need to delve into the details of their implementations, but customizability is not sacrificed and advanced users can inspect, modify, and extend practically every aspect of the package. Moreover, integration

---

[1]This manuscript refers to NETKET v3.5, released at the end of May 2022.

of our quantum object primitives with the JAX ecosystem allows users to easily define custom neural-network architectures and compute a range of quantum mechanical quantities, as well as their gradients, which are auto-generated through JAX's tracing-based approach. JAX provides the ability to write numerical code in pure Python using NUMPY-like calls for array operations, while still achieving high performance through just-in-time compilation using XLA, the accelerated linear algebra compiler that underlies TensorFlow. We have also integrated JAX and MPI with the help of MPI4JAX [38] to make NETKET scale to hundreds of computing nodes.

## 1.1 What's new

With the release of version 3, NETKET has moved from internally relying on a custom C++ core to the JAX framework, which allows models and algorithms to be written in pure Python and just-in-time compiled for high performance on both CPU and GPU platforms.[2] By using only Python, the installation process is greatly simplified and the barrier of entry for new contributors is lowered.

iFrom a user perspective, the most important new feature is the possibility of writing custom NQS wave functions using JAX, which allows for quick prototyping and deployment, frees users from having to manually implement gradients due to JAX's support for automatic differentiation, and makes models easily portable to GPU platforms. Other prominent new features are

- support for (real and imaginary time) unitary and Markovian dissipative dynamics;

- support for continuous systems;

- support for composite Hilbert spaces;

- efficient implementations of the quantum geometric tensor and stochastic reconfiguration, which scale to models with millions of parameters;

- group-invariant and group-equivariant layers and architectures which support arbitrary discrete symmetries.

A more advanced feature is an extension mechanism built around multiple dispatch [39], which allows users to override algorithms used internally by NETKET without editing the source itself. This can be used to make NETKET work with custom objects and algorithms to study novel problems that do not easily fit what is already available.

## 1.2 Outline

NETKET provides both an intuitive high-level interface with sensible defaults to welcome beginners, as well as a complete set of options and lower-level functions for flexible use by advanced users. The high-level interface is built around quantum-mechanical objects such as *Hilbert spaces* ( `netket.hilbert` ) and *operators* ( `netket.operator` ), presented in Section 2.

The central object in NETKET 3 is the *variational state,* discussed in Section 3, which bring together the neural-network ansatz, its variational parameters, and a Monte-Carlo sampler. In Section 3.2, we give an example on how to define an arbitrary neural network using a NETKET/JAX-compatible framework, while Section 3.4 presents the new API of

---

[2]Google's Tensor Processing Units (TPUs) are also, in principle, supported. However, at the time of writing they only support half-precision `float16` . Some modifications would be necessary to work-around loss of precision and gradient underflow.

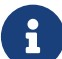 **JAX fluency.** Using NETKET's high-level interface and built-in neural network architectures does not require the user to be familiar with JAX and concepts such as just-in-time compilation and automatic differentiation. However, when defining custom classes such as neural network architectures, operators, or Monte Carlo samplers, some proficiency with writing JAX-compatible code will be required. We refrain from discussing JAX in detail and instead point the reader towards its documentation at jax.readthedocs.io.

stochastic samplers. In Section 3.5, we show how to compute the quantum geometric tensor (QGT) with NETKET, and compare the different implementations.

Section 4 shows how to use the three built-in optimization drivers to perform ground-state, steady-state, and dynamics calculations. Section 5 discusses NETKET's implementation of spatial symmetries and symmetric neural quantum states, which can be exploited to lower the size of the variational manifold and to target excited states in nontrivial symmetry sectors. In Section 6, we also show how to study a system with continuous degrees of freedom, such as interacting particles in one or more spatial dimensions.

The final sections present detailed workflow examples of some of the more common use cases of NETKET. In Section 7, we show how to study the ground state and the excited state of a lattice Hamiltonian. Section 9 gives examples of both unitary and Lindbladian dynamical simulations.

To conclude, Section 10 presents scaling benchmarks of NETKET running across multiple devices and a performance comparison with jVMC [40], another library similar in scope to NETKET.

Readers who are already familiar with the previous version of NETKET might be especially interested in the variational state interface described in Section 3.1, which replaces what was called *machine* in NETKET 2 [36], the QGT interface described in Section 3.5, algorithms for dynamics (Section 4.3 and Section 9), and symmetry-aware NQS (Section 5).

## 1.3 Installing NetKet

NETKET is a package written in pure Python; it requires a recent Python version, currently at least version 3.7. Even though NETKET itself is platform-agnostic, JAX, its main dependency, only works on MacOS and Linux at the time of writing.[3] Installing NETKET is straightforward and can be achieved running the following line inside a python environment:

```
1  pip install --upgrade netket
```

To enable GPU support, Linux with a recent CUDA version is required and a special version of JAX must be installed. As the appropriate installation procedure can change between JAX versions, we refer the reader to the official documentation[4] for detailed instructions.

NETKET by default does not make use of multiple CPUs that might be available to the user. Exploiting multiple processors, or even running across multiple nodes, requires MPI dependencies, which can be installed using the command

```
1  pip install --upgrade "netket[mpi]"
```

---

[3]In principle, JAX runs on Windows, but users must compile it themselves, which is not an easy process.
[4]https://github.com/google/jax#installation

These dependencies, namely `mpi4py` and `mpi4jax`, can only be installed if a working MPI distribution is already available.

Once NETKET is installed, it can be imported in a Python session or script and its version can be checked as

```
1  >>> import netket as nk
2  >>> print(nk.__version__)
3  3.5.0
```

We recommend that users use an up-to-date version when starting a new project. In code listings, we will often refer to the `netket` module as `nk` for brevity.

NETKET also comes with a set of so-called *experimental* functionalities which are packaged into the `netket.experimental` submodule which mirrors the structure of the standard `netket` module. Experimental APIs are marked as such because they are relatively young and we might want to change the function names or options keyword arguments without guaranteeing backward compatibility as we do for the rest of NETKET. In general, we import the experimental submodule as follows

```
1  >>> from netket import experimental as nkx
```

and use `nkx` as a shorthand for it.

## 2 Quantum-mechanical primitives

In general, when working with NETKET, the workflow is the following: first, one defines the Hilbert space of the system (Section 2.1) and the Hamiltonian or super-operator of interest (Section 2.2). Then, one builds a variational state (Section 3.1), usually combining a neural-network model and a stochastic sampler. In this section, we describe the first step in this process, namely, how to define a quantum-mechanical system to be modeled.

### 2.1 Hilbert spaces

Hilbert-space objects determine the state space of a quantum system and a specific choice of basis. Functionality related to Hilbert spaces is contained in the `nk.hilbert` module; for brevity, we will often leave out the prefix `nk.hilbert` in this section.

All implementations of Hilbert spaces derive from the class `AbstractHilbert` and fall into two classes:

- discrete Hilbert spaces, which inherit from the abstract class `DiscreteHilbert` and include spin (`Spin`), qubit (`Qubit`), Fock (`Fock`) as well as fermionic orbitals (`SpinOrbitalFermions`) Hilbert spaces. Discrete spaces are typically used to describe lattice systems. The lattice structure itself is, however, not part of the Hilbert space class and can be defined separately.

- continuous Hilbert spaces, which inherit from the abstract class `ContinuousHilbert`. Currently, the only concrete continuous space provided by NETKET is `Particle`.

Continuous Hilbert spaces are discussed in Section 6. A general discrete space with $N$ sites has the structure

$$\mathcal{H}_{\text{discrete}} = \text{span}\{|s_0\rangle \otimes \cdots \otimes |s_{N-1}\rangle \mid s_i \in \mathcal{L}_i, i \in \{0, \ldots, N-1\}\}, \tag{1}$$

where $\mathcal{L}_i$ is the set of local quantum numbers at site $i$ (e.g., $\mathcal{L} = \{0, 1\}$ for a qubit, $\mathcal{L} = \{\pm 1\}$ for a spin-1/2 system in the $\sigma^z$ basis, or $\mathcal{L} = \{0, 1, \ldots, N_{\max}\}$ for a Fock space with up to $N_{\max}$ particles per site). Constraints on the allowed quantum numbers are supported, resulting in Hilbert spaces that are subspaces of Eq. (1). For example, `Spin(1/2, total_sz=0)` creates a spin-1/2 space which only includes configurations $\{s_i\}$ that satisfy $\sum_{i=1}^{N} s_i = 0$. The corresponding basis states $|s\rangle$ span the zero-magnetization subspace. Similarly, constraints on the total population in Fock spaces are also supported.

Different spaces can be composed to create coupled systems by using the exponent operator ( `**` ) and the multiplication operator ( `*` ). For example, the code below creates the Hilbert space of a bosonic cavity with a cutoff of 10 particles at each site, coupled to 6 spin$-\frac{1}{2}$ degrees of freedom.

```
1  >>> hi = nk.hilbert.Fock(10) * nk.hilbert.Spin(1/2)**6
2  >>> print("Size of the hilbert space: ", hi.n_states)
3  Size of the hilbert space:  704
4  >>> print("Size of the basis: ", hi.size)
5  Size of the basis:  7
6  >>> hi.random_state(jax.random.PRNGKey(0), (2,))
7  DeviceArray([[10., -1.,  1., -1.,  1., -1., -1.],
8              [ 9.,  1., -1., -1.,  1., -1.,  1.]], dtype=float32)
```

All Hilbert objects can generate random basis elements through the function `random_state(rng_key, shape, dtype)`, which has the same signature as standard random number generators in JAX. The first argument is a JAX random-generator state as returned by `jax.random.PRNGKey`, while the other arguments specify the number of output states and optionally the JAX data type. In this example, an array with two state vectors has been returned. The first entry of each corresponds to the Fock space and is thus an integer in $\{0, 1, \ldots, 10\}$, while the rest contains the spin quantum numbers.

Custom Hilbert spaces can be constructed by defining a class inheriting either from `ContinuousHilbert` for continuous spaces or `DiscreteHilbert` for discrete spaces. In the rest of the paper, we will always be working with discrete Hilbert spaces unless stated otherwise.

NETKET also supports working with super-operators, such as the Liouvillian used to define open quantum systems, and variational mixed states. The density matrix is an element of the space of linear operators acting on a Hilbert space, $\mathbb{B}(\mathcal{H})$. NETKET represents this space using the Choi–Jamilkowski isomorphism [41, 42] convention $\mathbb{B}(\mathcal{H}) \sim \mathcal{H} \otimes \mathcal{H}$; this "doubled" Hilbert space is implemented as `DoubledHilbert`. Doubled Hilbert spaces behave largely similarly to standard Hilbert spaces, but their bases have double the number of degrees of freedom; for example, super-operators can be defined straightforwardly as operators acting on them.

## 2.2 Linear operators

NETKET is designed to allow users to work with large systems, beyond the typically small system sizes that are accessible through exact diagonalization techniques. In order to compute expectation values $\langle \hat{O} \rangle$ on such large spaces, we must be able to efficiently represent the operators $\hat{O}$ and work with their matrix elements $\langle \sigma | \hat{O} | \eta \rangle$ without storing them in memory.

NETKET provides different implementations for the operators, tailored for different use cases, which are available in the `netket.operator` submodule. NETKET operators are always defined relative to a specific underlying Hilbert space object and inherit from one of the abstract classes `DiscreteOperator` or `ContinuousOperator`, depending on the

classes of supported Hilbert spaces. We defer the discussion of operators acting on a continuous space to Section 6 and focus on discrete-space operators in the remainder of this section.

An operator acting on a discrete space can be represented as a matrix with some matrix elements $\langle \sigma | \hat{O} | \eta \rangle$. As most of those elements are zero in physical systems, a standard approach is to store the operator as sparse matrices, a format that lowers the memory cost by only storing non-zero entries. However, the number of non-zero matrix elements still scales exponentially with the number of degrees of freedom, so sparse matrices cannot scale to the thousands of lattice sites that we want to support, either. For this reason, NETKET uses one of three custom formats to represent operators:

- `LocalOperator` is an implementation that can efficiently represent sums of $K$-local operators, that is, operators that only act nontrivially on a set of $K$ sites. The memory cost of this format grows linearly with the number of operator terms and the number of degrees of freedom, but it scales exponentially in $K$.

- `PauliStrings` is an implementation that efficiently represents a product of Pauli $X, Y, Z$ operators acting on the whole system. This format only works with qubit-like Hilbert spaces, but it is extremely efficient and has negligible memory cost.

- `FermionOperator2nd` is an efficient implementation of second-quantized fermionic operators built out of the on-site creation and annihilation operators $f_i^\dagger, f_i$. It works together with `SpinOrbitalFermions` and the equivalent `Fock` spaces.

- Special implementations like `Ising`, which hard-code the matrix elements of the operator. Those are the most efficient, though they cannot be customized at all.

The `nk.operator` submodule also contains ready-made implementations of commonly used operators, such as Pauli matrices, bosonic ladder or projection operators, and common Hamiltonians such as the Heisenberg, or the Bose–Hubbard models.

### 2.2.1 Manipulating operators

Operators can be manipulated similarly to standard matrices: they can be added, subtracted, and multiplied using standard Python operators. In the example below we show how to construct the operator

$$\hat{O} = (\hat{\sigma}_0^x + \hat{\sigma}_1^x)^2 = 2(\hat{\sigma}_0^x \hat{\sigma}_1^x + 1). \tag{2}$$

starting from the Pauli $X$ operator acting on the $i$-th site, $\sigma_i^x$, given by the function `nk.operator.spin.sigmax(hi, i)`:

```
1  >>> hi = nk.hilbert.Spin(1/2)**2
2  >>> op = nk.operator.spin.sigmax(hi,0) + nk.operator.spin.sigmax(hi,1)
3  >>> op = op * op
4  >>> op
5  LocalOperator(dim=2, acting_on=[[0], [0, 1], [1]], constant=0, dtype=float64)
6  >>> op.to_dense()
7  array([[2., 0., 0., 2.],
8         [0., 2., 2., 0.],
9         [0., 2., 2., 0.],
10        [2., 0., 0., 2.]])
```

Note that each operator requires the Hilbert space object `hi` as well as the specific sites it acts on as constructor arguments. In the last step (line 6), we convert the operator into a dense matrix using the `to_dense()` method; it is also possible to convert an operator into a SciPy sparse matrix using `to_sparse()`.

While it is possible to inspect those operators and (if the Hilbert space is small enough) to convert them to dense matrices, NETKET's operators are built in order to support efficient row indexing, similar to row-sparse (CSR) matrices. Given a basis vector $|\sigma\rangle$ in a Hilbert space, one can efficiently query the list of basis states $|\eta\rangle$ and matrix elements $O(\sigma, \eta)$ such that

$$O(\sigma, \eta) = \langle\sigma|\hat{O}|\eta\rangle \neq 0 \tag{3}$$

using the function `operator.get_conn(sigma)`, which returns both the vector of non-zero matrix elements and the corresponding list of indices $|\eta\rangle$, stored as a matrix.[5]

# 3 Variational quantum states

In this section, we first introduce the general interface of variational states, which can be used to represent both pure states (vectors in the Hilbert space) and mixed states (positive-definite density operators). We then present how to define variational ansätze and the stochastic samplers needed that generate Monte Carlo states.

## 3.1 Abstract interface

A variational state describes a parametrized quantum state that depends on a (possibly large) set of variational parameters $\theta$. The quantum state can be either pure (denoted as $|\psi_\theta\rangle$) or mixed (written as a density matrix $\hat{\rho}_\theta$). NETKET defines an abstract interface, `netket.vqs.VariationalState`, for such objects; all classes that implement this interface will automatically work with all the high-level drivers (e.g., ground-state optimization or time-dependent variational dynamics) discussed in Section 4. The `VariationalState` interface is relatively simple, as it has only four requirements:

- The parameters $\theta$ of the variational state are exposed through the attribute `parameters` and should be stored as an array or a nested dictionary of arrays.

- The expectation value $\langle\hat{A}\rangle_\theta$ of an operator $\hat{A}$ can be computed or estimated by the method `expect`.

- The gradient of an expectation value with respect to the variational parameters, $\partial\langle\hat{A}\rangle_\theta/\partial\theta_j$, is computed by the method `expect_and_grad` [6].

- The quantum geometric tensor (Section 3.5) of a variational state can be constructed with the method `quantum_geometric_tensor`.

At the time of writing, NETKET exposes three types of variational state:

---

[5]This querying is currently performed in Python code, just-in-time compiled using NUMBA [43], which runs on the CPU. If you run your computations on a GPU with a small number of samples, this might introduce a considerable slowdown. We are aware of this issue and plan to adapt our operators to be indexed directly on the GPU in the future.

[6]For complex-valued parameters $\theta_j \in \mathbb{C}$, `expect_and_grad` returns the conjugate gradient $\partial\langle\hat{A}\rangle_\theta/\partial\theta_j^*$ instead. This is done because the conjugate gradient corresponds to the direction of steepest ascent when optimizing a real-valued function of complex variables [44].

- `nk.vqs.ExactState` represents a variational pure state $|\psi_\theta\rangle$ and computes expectation values, gradients and the geometric tensor by performing exact summation over the full Hilbert space.

- `nk.vqs.MCState` (short for *Monte Carlo state*) represents a variational pure state and computes expectation values, gradients and the geometric tensor by performing Markov chain Monte Carlo (MCMC) sampling over the Hilbert space.

- `nk.vqs.MCMixedState` represents a variational mixed state and computes expectation values by sampling diagonal entries of the density matrix.

Variational states based on Monte Carlo sampling are the main tools that we expose to users, together with a wide variety of high-performance Monte Carlo samplers. More details about stochastic estimates and Monte Carlo sampling will be discussed in Section 3.3 and Section 3.4.

**Dispatch and algorithm selection.** With three different types of variational state and several different operators supported, it is hard to write a well-performing algorithm that works with all possible combinations of types that users might require. In order not to sacrifice performance for generic algorithms, NETKET uses the approach of multiple dispatch based on the PLUM module [39]. Combined with JAX's just-in-time compilation, this solution bears a strong resemblance to the approach commonly used in the Julia language [6].

Every time the user calls `VariationalState.expect` or `.expect_and_grad`, the types of the variational state and the operator are used to select the most specific algorithm that applies to those two types. This allows NETKET to provide generic algorithms that work for all operators, but keeps it easy to supply custom algorithms for specific operator types if desired.

This mechanism is also exposed to users: it is possible to override the algorithms used by NETKET to compute expectation values and gradients without modifying the source code of NETKET but simply by defining new dispatch rules using the syntax shown below.

```
1  @nk.vqs.expect.dispatch
2  def expect(vstate: MCState, operator: Ising):
3      # more efficient implementation than default one
4      #
5      # expectation_value = ...
6      #
7      return expectation_value
```

## 3.2 Defining the variational ansatz

The main feature defining a variational state is the parameter-dependent mapping of an input configuration to the corresponding probability amplitude or, in other words, the quantum wave function (for pure states)

$$(\theta, s) \mapsto \psi_\theta(s) = \langle s|\psi_\theta\rangle \tag{4}$$

or quantum density matrix (for mixed states)

$$(\theta, s, s') \mapsto \rho_\theta(s, s') = \langle s|\, \hat{\rho}_\theta\, |s'\rangle. \tag{5}$$

In NETKET, this mapping is called a *model* (of the quantum state).[7] In the case of NQS, the model is given by a neural network. For defining models, NETKET primarily relies on FLAX [45], a JAX-based neural-network library[8]. Ansätze are implemented as FLAX modules that map input configurations (the structure of which is determined by the Hilbert space) to the corresponding log-probability amplitudes. For example, pure quantum states are evaluated as

$$\ln \psi_\theta(s) = \texttt{module.apply}\ (\theta, s). \tag{6}$$

The use of log-amplitudes has the benefit that the log-derivatives $\partial \ln \psi_\theta(s)/\partial \theta_j$, often needed in variational optimization algorithms, are directly available through automatic differentiation of the model. It also makes it easier for the model to learn amplitudes with absolute values ranging over several orders of magnitude, which is common for many types of quantum states.

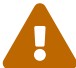

**Real and complex amplitudes.** NETKET supports both real-valued and complex-valued model outputs. However, since model outputs correspond to log-amplitudes, real-valued networks can only represent states that have exclusively non-negative amplitudes, $\ln \psi_\theta(s) \in \mathbb{R} \Rightarrow \psi_\theta(s) \geq 0$.
Since the input configurations $s$ are real, in many pre-defined NETKET models the data type of the network parameters ($\theta \in \mathbb{R}^{N_\mathrm{P}}$ or $\theta \in \mathbb{C}^{N_\mathrm{P}}$) determines whether an ansatz represents a general or a real non-negative state. This should be kept in mind in particular when optimizing Hamiltonians with ground states that can have negative amplitudes.

### 3.2.1 Custom models using Flax

The recommended way to define a custom module is to subclass `flax.linen.Module` and to provide a custom implementation of the `__call__` method. As an example, we define a simple one-layer NQS with a wave function of the form

$$\ln \psi(s) = \sum_{j=1}^{M} \tanh[Ws + b]_j. \tag{7}$$

with the number of visible units $N$ matching the number of physical sites, a number of hidden units $M$, and complex parameters $W \in \mathbb{C}^{M \times N}$ (the weight matrix) and $b \in \mathbb{C}^M$ (the bias vector). Using NETKET and FLAX, this ansatz can be implemented as follows:

```
1  import netket as nk
2  import jax.numpy as jnp
3  import flax
4  import flax.linen as nn
5
```

---

[7]The notion of "model" in NETKET 3 is related to the "machine" classes in NETKET 2 [36]. However, while NETKET 2 machines both define the mapping (4) and store the current parameters, this has been decoupled in NETKET 3. The model only specifies the mapping, while the parameters are stored in the variational state classes.

[8]While our primary focus has been the support of FLAX, NETKET can in principle be used with *any* JAX-compatible neural network model. For example, NETKET currently includes a compatibility layer which ensures that models defined using the HAIKU framework by DeepMind [46] will work automatically as well. Furthermore, any model represented by a pair of `init` and `apply` functions (as used, e.g., in the STAX framework included with JAX) is also supported.

```
 6   class OneLayerNQS(nn.Module):
 7      # Module hyperparameter:
 8      n_hidden_units: int
 9
10      @nn.compact
11      def __call__(self, s):
12          n_visible_units = s.shape[-1]
13          # define parameters
14          # the arguments are: name, initializer, shape, dtype
15          W = self.param(
16              "weights",
17              nn.initializers.normal(),
18              (self.n_hidden_units, n_visible_units),
19              jnp.complex128,
20          )
21          b = self.param(
22              "bias",
23              nn.initializers.normal(),
24              (self.n_hidden_units,),
25              jnp.complex128,
26          )
27
28          # multiply with weight matrix over last dimension of s
29          y = jnp.einsum("ij,...j", W, s)
30          # add bias
31          y += b
32          # apply tanh activation and sum
33          y = jnp.sum(jnp.tanh(y), axis=-1)
34
35          return y
```

The decorator `flax.linen.compact` used on `__call__` (line 10) makes it possible to define the network parameters directly in the body of the call function via `self.param` as done above (lines 15 and 21). For performance reasons, the input to the module is batched. This means that, instead of passing a single array of quantum numbers $s$ of size $N$, a batch of multiple state vectors is passed as a matrix of shape `(batch_size, N)`. Therefore, operations like the sum over all feature indices in the example above need to be explicitly performed over the last axis.

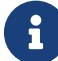

> **Just-in-time compilation.** Note that the network will be just-in-time (JIT) compiled to efficient machine code for the target device (CPU, GPU, or TPU) using `jax.jit`, which means that all code inside the `__call__` method needs to written in a way compatible with `jax.jit`.
>
> In particular, users should use `jax.numpy` for NumPy calls that need to happen at runtime; explicit Python control flow, such as `for` loops and `if` statements, should also be avoided, unless one explicitly wants to have them evaluate once at compile time. We refer users to the JAX documentation for further information on how to write efficient JIT-compatible code.

The module defined above can be used by first initializing the parameters using `module.init` and then computing log-amplitudes through `module.apply`:

```
 1   >>> module = OneLayerNQS(n_hidden_units=16)
```

```
2   # init takes two arguments, a PRNG key for random initialization
3   # and a dummy array used to determine the input shape
4   # (here with a batch size of one):
5   >>> params = module.init(nk.jax.PRNGKey(0), jnp.zeros((1, 8)))
6   >>> module.apply(params, jnp.array([[-1, 1, -1, 1, -1, 1, -1, 1]]))
7   DeviceArray([-0.00047843+0.07939122j], dtype=complex128)
```

### 3.2.2   Network parametrization and pytrees

**Parameter data types.** NETKET supports models with both real-valued and complex-valued network parameters. The data type of the parameters does not determine the output type. It is possible to define a model with real parameters that produces complex output. A simple example is the sum of two real-valued and real-parameter networks, representing real and imaginary part of the log-amplitudes (and thus phase and absolute value of the wave function) [30, 32]:

$$\ln \psi_{(\theta,\eta)}(s) = f_\theta(s) + i g_\eta(s) \quad \Leftrightarrow \quad |\psi_{(\theta,\eta)}(s)| = \exp[f_\theta(s)], \quad \arg \psi_{(\theta,\eta)}(s) = g_\eta(s) \quad (8)$$

(where all $\theta_i, \eta_i \in \mathbb{R}$ and $f(s), g(s) \in \mathbb{R}$).

More generally, any model with $N_\mathrm{p}$ complex parameters $\theta = \alpha + i\beta$ can be represented by a model with $2N_\mathrm{p}$ real parameters $(\alpha, \beta)$. While these parametrizations are formally equivalent, the choice of complex parameters can be particularly useful in the case where the variational mapping is holomorphic or, equivalently, complex differentiable with respect to $\theta$. This is the case for many standard network architectures such as RBMs or feed-forward networks, since both linear transformations and typical activation functions are holomorphic (such as tanh, cosh, and their logarithms) or piecewise holomorphic (such as ReLU), which is sufficient in practice. Note, however, that there are also common architectures, such as autoregressive networks, that are not holomorphic. In the holomorphic case, the computational cost of differentiating the model, e.g., to compute the quantum geometric tensor (Section 3.5), can be reduced by exploiting the Cauchy–Riemann equations [47],

$$i\nabla_\alpha \psi_{(\alpha,\beta)}(s) = \nabla_\beta \psi_{(\alpha,\beta)}(s). \quad (9)$$

Note that NETKET generally supports models with *arbitrary parametrizations* (i.e., real and both holomorphic and non-holomorphic complex parametrizations). The default assumption is that models with complex weights are non-holomorphic, but some objects (most notably the quantum geometric tensor) accept a flag `holomorphic=True` to enable a more efficient code path for holomorphic networks.

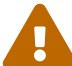 It is the user's responsibility to only set `holomorphic=True` for models that are, in fact, holomorphic. If this is incorrectly specified, NETKET code may give incorrect results. To check whether a specific architecture is holomorphic, one can verify the condition

$$\partial\psi_\theta(s)/\partial\theta_j^* = (\partial/\partial\alpha_j + i\partial/\partial\beta_j)\psi_\theta(s) = 0, \quad (10)$$

which is equivalent to Eq. (9).

**Pytrees.** In NETKET, model parameters do not need to be stored as a contiguous vector. Instead, models can support any collection of parameters that forms a so-called *pytree*.

Pytree is JAX terminology for collections of numerical tensors stored as the leaf nodes inside layers of nested standard Python containers (such as lists, tuples, and dictionaries).[9] Any object that is not itself a pytree, in particular NumPy or JAX arrays, is referred to as a *leaf*. Networks defined as FLAX modules store their parameters in a (potentially nested) dictionary, which provides name-based access to the network parameters.[10] For the `OneLayerNQS` defined above, the parameter pytree has the structure:

```
1  # For readability, the actual array data has been replaced with ... below.
2  >>> print(params)
3  FrozenDict({
4      params: {
5          weights: DeviceArray(..., dtype=complex128),
6          bias: DeviceArray(..., dtype=complex128),
7      },
8  })
```

The names of the entries in the parameter dictionary correspond to those given in the `param` call when defining the model. NETKET functions often work directly with both plain arrays and pytrees of arrays. Furthermore, any Python function can be applied to the leaves of a pytree using `jax.tree_map`. For example, the following code prints a pytree containing the shape of each leaf of `params`, preserving the nested dictionary structure:

```
1  >>> print(jax.tree_map(jnp.shape, params))
2  FrozenDict({
3      params: {
4          weights: (16, 8),
5          bias: (16,),
6      },
7  })
```

Functions accepting multiple leaves as arguments can be mapped over the corresponding number of pytrees (with compatible structure) using `jax.tree_map`. For example, the difference of two parameter pytrees of the same model can be computed using `delta = jax.tree_map(lambda a, b: a - b, params1, params2)`. NETKET provides an additional set of utility functions to perform linear algebra operations on such pytrees in the `nk.jax` submodule.

### 3.2.3   Pre-defined ansätze included with NetKet

NETKET provides a collection of pre-defined modules under `nk.models`, which allow quick access to many commonly used NQS architectures (Table 1):

- **Jastrow:** The Jastrow ansatz [48,49] is an extremely simple yet effective many-body ansatz that can capture some inter-particle correlations. The log-wavefunction is the linear function $\log\psi(\sigma) = \sum_i \sigma_i W_{i,j} \sigma_j$. Evaluation of this ansatz is very fast but it is also the least powerful model implemented in NETKET;

---

[9]See https://github.com/google/jax/blob/jax-v0.2.28/docs/pytrees.md for a detailed introduction of pytrees.

[10]Specifically, FLAX stores networks parameters in an immutable `FrozenDict` object, which otherwise has the same semantics as a standard Python dictionary and, in particular, is also a valid pytree. The parameters can be modified by converting to a standard mutable `dict` via `flax.core.unfreeze(params)`.

| Name | NETKET class | References |
|---|---|---|
| Jastrow ansatz | `Jastrow` | [48,49] |
| Restricted Boltzmann machine (RBM) | `RBM` , `RBMMultiVal` , `RBMModPhase` | [8] |
| Symmetric RBM | `RBMSymm` | [8] |
| Group-Equivariant Convolutional Neural Network | `GCNN` | Section 5 [50] |
| Autoregressive Neural Network | `ARNNDense` , `ARNNConv1D` , `ARNNConv2D` , `FastARNNConv1D` , `FastARNNConv2D` | [51] |
| Neural Density Matrix | `NDM` | [25,52] |

Table 1: List of models included in NETKET's `nk.models` submodule, together with relevant references.

- **RBM:** The restricted Boltzmann machine (RBM) ansatz is composed by a dense layer followed by a nonlinearity. If the Hilbert space has $N$ degrees of freedom of size $d$, `RBM` has $\alpha N$ features in the dense layer. This ansatz requires `param_dtype=complex` to represent states that are non-positive valued. `RBMMultiVal` is a one-hot encoding layer followed by an `RBM` with $\alpha d N$ features in its dense layer. Finally, `RBMModPhase` consists of two real-valued RBMs that encode respectively the modulus and phase of the wavefunction as $\log \psi(\sigma) = $ `RBM` $(\sigma) + i$ `RBM` $(\sigma)$. This ansatz only supports real parameters. If considering Hilbert spaces with local dimension $d > 2$, plain RBMs usually require a very large feature density $\alpha$ and `RBMMultiVal` s perform better.

- **RBMSymm:** A symmetry-invariant RBM. Only symmetry groups that can be represented as permutations of the computational basis are supported (see Section 5). This architecture has fewer parameters than an RBM, but it is more expensive to evaluate. It requires `param_dtype=complex` to represent states that are non-positive valued.

- **GCNN:** A symmetry-equivariant feed-forward network (see Section 5.2). Only symmetry groups that can be represented as permutations of the computational basis are supported. This model is much more complex and computationally intensive than `RBMSymm` , but can also lead to more accurate results. It can also be used to target an excited state of a lattice Hamiltonian. When working with states that are real but non-positive, one can use real parameters together with `complex_output=True` . If the states are to have a complex phase, `param_dtype=complex` is required.

- **Autoregressive networks:** ARNNs are models that can produce uncorrelated samples when sampled with `nk.sampler.ARNNSampler` . Those architectures can be efficiently sampled on GPUs, but they are much more expensive than traditional RBMs.

- **NDM:** A positive-semidefinite density matrix ansatz, comprised of a component describing the pure part and one describing the mixed part of the state. The pure part is equivalent to an RBM with feature density $\alpha$, while the mixed part is an RBM with feature density $\beta$. This network only supports real parameters.

### 3.2.4   Custom layers included with Flax and NetKet

The `nk.nn` submodule contains generic modules such as masked dense, masked convolutional and symmetric layers to be used as building blocks for custom neural networks. Those layers are complementary to those provided by FLAX and can be combined together to develop novel neural-network architectures.[11]

As an example, a multi-layer NQS with two convolutional and one final dense layer acting as a weighted sum can be defined as follows:

```python
class MultiLayerCNN(nn.Module):
    features1: int
    features2: int
    kernel_size: int

    @nn.compact
    def __call__(self, s):
        # define layers
        layer1 = nn.Conv(
            features=self.features1,
            kernel_size=self.kernel_size,
        )
        layer2 = nn.Conv(
            features=self.features2,
            kernel_size=self.kernel_size,
        )
        weighted_sum = nn.Dense(features=1)

        # apply layers and tanh activations
        y = jnp.tanh(layer1(s))
        y = jnp.tanh(layer2(y))
        y = weighted_sum(y)
        # last axis only has one entry, so we just return that
        # but keep the batch dimension
        return y[..., 0]
```

FLAX network layers are available from the `flax.linen` submodule (imported as `nn` in the example above), NETKET layers from `netket.nn`.

### 3.3   Estimating observables

For any variational ansatz, it is crucial to also have efficient algorithms for computing quantities of interest, in particular observables and their gradients. Since evaluating the wave function on all configurations is infeasible for larger Hilbert spaces, NQS approaches rely on Monte Carlo sampling of quantum expectation values.

---

[11]In the past FLAX had minor issues with complex numbers and therefore NETKET included versions of some standard layers, such as `Dense` and `Conv`, that handle complex numbers properly. Starting with FLAX version 0.5, released in May 2022, those issues have been addressed and we now recommend the use of FLAX layers also with complex numbers.

**Pure states.** The quantum expectation value of an operator $\hat{A}$ on a non-normalized pure state $|\psi\rangle$ can be written as a classical expectation value over the Born distribution $p(s) \propto |\psi(s)|^2$ using the identity

$$\langle \hat{A} \rangle = \frac{\langle\psi|\hat{A}|\psi\rangle}{\langle\psi|\psi\rangle} = \sum_s \frac{|\psi(s)|^2}{\langle\psi|\psi\rangle} \tilde{A}(s) = \sum_s p(s)\tilde{A}(s) = \mathbb{E}[\tilde{A}], \tag{11}$$

where $\tilde{A}$ is the *local estimator*

$$\tilde{A}(s) = \frac{\langle s|\hat{A}|\psi\rangle}{\langle s|\psi\rangle} = \sum_{s'} \frac{\psi(s')}{\psi(s)}\langle s|\hat{A}|s'\rangle, \tag{12}$$

also known as the *local energy* when $\hat{A}$ is the Hamiltonian [53]. Even though the sum in Eq. (12) runs over the full Hilbert space basis, the local estimator can be efficiently computed if the operator is sufficiently sparse in the given basis, i.e., all but a tractable number of matrix elements $\langle s|\hat{A}|s'\rangle$ are zero. Thus, an efficient algorithm is required that, given $s$, yields all connected configurations $s'$ together with their respective matrix elements, as described in Section 2.2. Given the derivatives of the log-amplitudes

$$O_i(s) = \frac{\partial \ln \psi_\theta(s)}{\partial \theta_i}, \tag{13}$$

gradients of expectation values can also be evaluated. Define the *force vector* as the covariance

$$f_i = \mathrm{Cov}[O_i, \tilde{A}] = \mathbb{E}[O_i^*(\tilde{A} - \mathbb{E}[\tilde{A}])]. \tag{14}$$

Then, if $\theta_i \in \mathbb{R}$ is a real-valued parameter,

$$\frac{\partial\langle\hat{A}\rangle}{\partial\theta_i} = 2\,\mathrm{Re}[f_i]. \tag{15}$$

If $\theta_i \in \mathbb{C}$ and the mapping $\theta_i \mapsto \psi_\theta(s)$ is complex differentiable (holomorphic),

$$\frac{\partial\langle\hat{A}\rangle}{\partial\theta_i^*} = f_i. \tag{16}$$

In case of a non-holomorphic mapping, $\mathrm{Re}[\theta_i]$ and $\mathrm{Im}[\theta_i]$ can be treated as two independent real parameters and Eq. (15) applies to each.

The required classical expectation values are then estimated by averaging over a sequence $\{s_i\}_{i=1}^{N_\mathrm{s}}$ of configurations distributed according to the Born distribution $p(s) \propto |\psi(s)|^2$; e.g., Eq. (11) becomes

$$\mathbb{E}[\tilde{A}] \approx \frac{1}{N_\mathrm{s}} \sum_{i=1}^{N_\mathrm{s}} \tilde{A}(s). \tag{17}$$

For some models, in particular autoregressive neural networks [51], one can efficiently draw samples from the Born distribution directly. For a general ansatz, however, this is not possible and Markov-chain Monte Carlo (MCMC) sampling methods [53] must be used: these generate a sequence (Markov chain) of samples that asymptotically follows the Born distribution. Such a chain can be generated using the Metropolis–Hastings algorithm [54], which is implemented in NETKET's sampler interface, described in the next section.

**Mixed states.** When evaluating observables for mixed states, it is possible to exploit a slightly different identity,

$$\langle \hat{A} \rangle = \frac{\text{Tr}\left[\hat{\rho}\hat{A}\right]}{\text{Tr}[\hat{\rho}]} = \sum_s \frac{\rho(s,s)}{\text{Tr}[\hat{\rho}]} \tilde{A}_\rho(s) = \mathbb{E}[\tilde{A}_\rho], \tag{18}$$

which rewrites the quantum expectation value as a classical expectation over the probability distribution defined by the diagonal of the density matrix $p(s) \propto \rho(s,s)$. Here, $\tilde{A}_\rho$ denotes the local estimator of the observable *over a mixed state,*

$$\tilde{A}_\rho(s) = \frac{\langle s|\hat{\rho}\hat{A}|s\rangle}{\langle s|\,\hat{\rho}\,|s\rangle} = \sum_{s'} \frac{\rho(s,s')}{\rho(s,s)} \langle s'|\hat{A}|s\rangle. \tag{19}$$

It is then possible to follow the same procedures detailed in the previous paragraph for pure states to compute the gradient of an expectation value of an operator over a mixed state by replacing the probability distribution over which the average is computed and the local estimator.

### 3.3.1 Reducing memory usage with chunking

The number of variational state evaluations required to compute the local estimators (12) typically scales superlinearly[12] in the number of sites $N$. For optimal performance, NETKET by default performs those evaluations in a single call using batched inputs. However, for large Hilbert spaces or very deep models it might be impossible to fit all required intermediate buffers into the available memory, leading to out-of-memory errors. This is encountered particularly often in calculations on GPUs, which have more limited memory.

To avoid those errors, NETKET's `nk.vqs.VariationalState` exposes an attribute called `chunk_size`, which controls the maximum number of configurations for which a model is evaluated at the same time[13]. The chunk size effectively bounds the maximum amount of memory required to evaluate the variational function at the expense of an increased computational cost in some operations involving the derivatives of the model. For this reason, we suggest using the largest chunk size that fits in memory.

Chunking is supported for the majority of operations, such as computing expectation values and their gradients, as well as the evaluation the quantum geometric tensor. If a chunk size is specified but an operation does not support it, NETKET will print a warning and attempt to perform the operation without chunking.

## 3.4 Monte Carlo samplers

The sampling algorithm used to obtain a sequence of configurations from the probability distribution defined by the variational ansatz is specified by sampler classes inheriting from `nk.sampler.AbstractSampler`. Following the purely functional design of JAX, we define the sampler to be a stateless collection of settings and parameters, while storing all mutable state such as the PRNG key and the statistics of acceptances in an immutable sampler state object. Both the sampler and the sampler state are stored in the variational state, but they can be used independently, as they are decoupled from the rest of NETKET.

---

[12]The exact scaling depends on the sparsity of the observable in the computational basis (which in a lattice model primarily depends on the locality of the operator and the dimension of the lattice).

[13]The chunk size can be specified at model construction and freely changed later. Chunking can also be disabled at any time by setting `VariationalState.chunk_size = None`.

The Metropolis–Hastings algorithm is used to generate samples from an arbitrary probability distribution. In each step, it suggests a transition from the current configuration $s$ to a proposed configuration $s'$. The proposal is accepted with probability

$$P_{\text{acc}}(s \rightarrow s') = \min\left(1, \frac{P(s')}{P(s)}\frac{g(s \mid s')}{g(s' \mid s)}\right), \tag{20}$$

where $P$ is the distribution being sampled from and $g(s' \mid s)$ is the conditional probability of proposing $s'$ given the current $s$. We use $L(s, s') = \log[g(s \mid s')/g(s' \mid s)]$ to denote the correcting factor to the log probability due to the transition kernel. This factor is needed for asymmetric kernels that might propose one move with higher probability than its reverse. Simple kernels, such as a local spin flip or exchange, are symmetric, therefore $L(s, s') = L(s', s) = 1$, but other proposals, such as Hamiltonian sampling, are not necessarily symmetric and need this factor.

At the time of writing, NETKET exposes four types of rules to use with the Metropolis sampler: `MetropolisLocal`, which changes one discrete local degree of freedom in each transition; `MetropolisExchange`, which exchanges two local degrees of freedom respecting a conserved quantity (e.g., total particle number or magnetization); `MetropolisHamiltonian`, which transitions the configuration according to the off-diagonal elements of the Hamiltonian; and `MetropolisGaussian`, which moves a configuration with continuous degrees of freedom according to a Gaussian distribution.

The different transition kernels in these samplers are represented by `MetropolisRule` objects. To define a Metropolis sampling algorithm with a new transition kernel, one only needs to subclass `MetropolisRule` and implement the `transition` method, which gives $s'$ and $L(s, s')$ in each transition. For example, the following transition rule changes the local degree of freedom on two sites at a time:

```python
from netket.hilbert.random import flip_state
from netket.sampler import MetropolisRule
from netket.utils.struct import dataclass

# To be jax-compatible, it must be a dataclass
@dataclass
class TwoLocalRule(MetropolisRule):
    def transition(rule, sampler, machine, parameters, state, key, σ):
        # Deduce the number of MCMC chains from input shape
        n_chains = σ.shape[0]
        # Load the Hilbert space of the sampler
        hilb = sampler.hilbert
        # Split the rng key into 2: one for each random operation
        key_indx, key_flip = jax.random.split(key, 2)
        # Pick two random sites on every chain
        indxs = jax.random.randint(
            key_indx, shape=(n_chains, 2), minval=0, maxval=hilb.size
        )
        # flip those sites
        σp, _ = flip_state(hilb, key_flip, σ, indxs)

        # If this transition had a correcting factor L, it's possible
        # to return it as a vector in the second value
        return σp, None
```

Once a custom rule is defined, a MCMC sampler using such rule can be constructed with the command `sampler = MetropolisSampler(hilbert, TwoLocalRule())`. Besides

| Type | Name | Usage |
|---|---|---|
| MCMC (Metropolis) | `MetropolisLocal` | discrete Hilbert spaces |
| | `MetropolisExchange` | permutations of local states, conserving total magnetization in spin systems |
| | `MetropolisHamiltonian` | preserving symmetries of the Hamiltonian |
| | `MetropolisGaussian` | continuous Hilbert spaces |
| Direct | `ExactSampler` | small Hilbert spaces, performs MC sampling from the exact distribution |
| | `ARDirectSampler` | autoregressive models |

Table 2: List of samplers in NETKET with their class names and a description.

Metropolis algorithms, more advanced Markov chain algorithms can also be implemented as NETKET samplers. Currently, parallel tempering is provided as an experimental feature.

Some models allow us to directly generate samples that are exactly distributed according to the desired probability, without the use of Markov chains and the issue of autocorrelation, which often leads to more efficient sampling. In this case, direct samplers can be implemented with an interface similar to Markov chain samplers. Currently NETKET has implemented `ARDirectSampler` to be used with ARNNs. For benchmarking purposes, NETKET also provides `ExactSampler`, which allows direct sampling from any model by computing the full Born distribution $|\psi(s)|^2$ for all $s$. Table 2 is a list of all the samplers.

## 3.5 Quantum geometric tensor

The quantum geometric tensor (QGT) [55] of a pure state is the metric tensor induced by the Fubini–Study distance [56,57]

$$d(\psi, \phi) = \cos^{-1} \sqrt{\frac{\langle\psi|\phi\rangle\langle\phi|\psi\rangle}{\langle\psi|\psi\rangle\langle\phi|\phi\rangle}}, \qquad (21)$$

which is the natural and gauge-invariant distance between two pure quantum states $|\psi\rangle$ and $|\phi\rangle$. The QGT is commonly used for time evolution (see Section 4.3) and for quantum natural gradient descent [58], which was originally developed in the VMC community under the name of *stochastic reconfiguration* (SR) [59]. Quantum natural gradient descent is directly related to the natural gradient descent developed in the machine learning community [60].

From now on, we assume that the state $|\psi_\theta\rangle$ is parametrized by a set of parameters $\theta$. Assuming further that $|\phi\rangle = |\psi_{\theta+\delta\theta}\rangle$, the distance (21) can be expanded to second order in the infinitesimal parameter change $\delta\theta$ as $d(\psi_\theta, \psi_{\theta+\delta\theta})^2 = (\delta\theta)^\dagger G(\delta\theta)$, where $G$ is the quantum geometric tensor. For a holomorphic mapping $\theta \mapsto |\psi_\theta\rangle$, the QGT is given by

$$G_{ij}(\theta) = \frac{\left\langle \partial_{\theta_i}\psi_\theta \middle| \partial_{\theta_j}\psi_\theta \right\rangle}{\langle\psi|\psi\rangle} - \frac{\left\langle \partial_{\theta_i}\psi_\theta \middle| \psi_\theta \right\rangle \left\langle \psi_\theta \middle| \partial_{\theta_j}\psi_\theta \right\rangle}{\langle\psi|\psi\rangle^2} \qquad (22)$$

where the indices $i, j$ label the parameters and $\langle x | \partial_{\theta_i} \psi_\theta \rangle = \partial_{\theta_i} \langle x | \psi_\theta \rangle$. Similar to expectation values and their gradients, Eq. (22) can be rewritten as a classical covariance with respect to the Born distribution $\propto |\psi(s)|^2$:

$$G_{ij}(\theta) = \mathrm{Cov}[O_i, O_j] = \mathbb{E}\left[O_i^*(O_j - \mathbb{E}[O_j])\right], \tag{23}$$

where $O_i$ are the log-derivatives (13) of the ansatz.[14] This allows the quantum geometric tensor to be estimated using the same sampling procedure used to obtain expectation values and gradients. The QGT or its stochastic estimate is also commonly known as the *S matrix* [53] or *quantum Fisher matrix* (QFM) in analogy to the classical Fisher information matrix [58, 60, 61].

For applications such as quantum natural gradient descent or time-evolution it is usually not necessary to access the full, dense matrix representation $G_{ij}(\theta)$ of the quantum geometric tensor, but only to compute its product with a vector, $\tilde{v}_i = \sum_j G_{ij}(\theta)\, v_j$. When the variational ansatz $\psi_\theta$ has millions of parameters, the QGT can indeed be too large to be stored in memory. Exploiting the Gram matrix structure of the geometric tensor [62], we can directly compute its action on a vector without ever calculating the full matrix, trading memory requirements for an increased computational cost.

Given a variational state `vs`, a QGT object can be obtained by calling:

```
1  >>> qgt = vs.quantum_geometric_tensor()
```

This `qgt` object does not store the full matrix, but can still be applied to a vector with the same shape as the parameters:

```
1  >>> vec = jax.tree_map(jnp.ones_like, vs.parameters)
2  >>> qgt_times_vec = qgt @ vec
```

It can be converted to a dense matrix by calling `to_dense`:

```
1  # get the matrix (2d array) of the qgt
2  >>> qgt_dense = qgt.to_dense()
3  # flatten vec into a 1d Array
4  >>> grad_dense, unravel = nk.jax.tree_ravel(vec)
5  >>> qgt_times_vec = unravel(qgt_dense @ vec_dense)
```

The QGT can then be used together with a direct solver, such as `jnp.linalg.eigh`, `jnp.linalg.svd`, or `jnp.linalg.qr`.

**Mixed states.** When working with mixed states, which are encoded in a density matrix, it is necessary to pick a suitable metric to induce the QGT. Even if the most physical distances for density matrices are the spectral norm or other trace-based norms [63, 64], it is generally hard to use them to define an expression for the QGT that can be efficiently sampled and computed at polynomial cost. While this might be regarded as a *barbaric* choice, it leads to an expression equivalent to Eq. (23), where the expectation value is over the joint-distribution of the of row and column labels $(s, s')$ of the squared density matrix $\propto |\rho(s, s')|^2$. Therefore, when working with mixed states, we resort to the $L_2$ norm, which is equivalent to treating the density matrix as a vector ("pure state") in an enlarged Hilbert space.

---

[14] Strictly speaking, this estimator is only correct if $\psi_\theta(s) = 0 \implies \partial_{\theta_k} \psi_\theta(s) = 0$. This is because we multiplied and divided by $\psi_\theta(s)$ in the derivation of the estimator, which is only valid if $\psi_\theta(s) \neq 0$.

### 3.5.1 Implementation differences

There is some freedom in the way one can calculate the QGT, each different implementation taking a different tradeoff between computational and memory cost. In the example above, we have relied on NETKET to automatically select the best implementation according to some internal heuristics, but if one wants to push variational methods to the limits, it is useful to understand the two different implementations on offer:

- `QGTJacobian`, which computes and stores the Jacobian $O_i(s)$ of the model (cf. Appendix B.1) at initialization (using reverse-mode automatic differentiation) and can be applied to a vector by performing two matrix-vector multiplications;

- `QGTOnTheFly`, which lazily computes the application of the quantum geometric tensor on a vector through automatic differentiation.

**QGTOnTheFly** is the most flexible and can be scaled to arbitrarily large systems. It is based on the observation that the QGT is the Jacobian of the model multiplied with its conjugate, which means that its action can be calculated by combining forward and reverse-mode automatic differentiation. At initialization, it only computes the linearization (forward pass), and then it effectively recomputes the gradients every time it is applied to a vector. However, since it never has to store these gradients, it is not limited by the available memory, which also makes it perform well for shallow neural-network models like RBMs. This method works with both holomorphic and non-holomorphic ansätze with no difference in performance.

**QGTJacobian.** For deep networks with ill-conditioned[15] quantum geometric tensors, recomputing the gradients at every step in an iterative solver might be very costly. `QGTJacobian` can therefore achieve better performance at the cost of considerably higher memory requirements because it precomputes the Jacobian at construction and stores it. The downside is that it has to store a *matrix* of shape $N_{\text{samples}} \times N_{\text{parameters}}$, which might not fit in the memory of a GPU. We note that there are two different implementations of `QGTJacobian`: `QGTJacobianDense` and `QGTJacobianPyTree`. The difference among the two is that in the former the Jacobian is stored contiguously in memory, leading to a better throughput on GPUs, while the latter stores them in the same structure as the parameters (so each parameter block is separated from the others). Converting from the non-contiguous (`QGTJacobianPyTree`) to the contiguous (`QGTJacobianDense`) format has, however, a computational and memory cost which might shadow its benefit. Moreover, the dense format does not work with non-homogeneous parameter data types. The basic `QGTJacobian` algorithm supports both holomorphic and non-holomorphic NQS, but a better performing algorithm for holomorphic ansätze can be accessed instantiating it with the option `holomorphic=True`.

Key differences between the different QGT implementations are summarized in Table 3. Implementations can be selected, and options passed to them, as shown below:

```
1  >>> from netket.optimizer.qgt import (
2  ...     QGTJacobianPyTree, QGTJacobianDense, QGTOnTheFly
3  ... )
4  >>> qgt1 = vs.quantum_geometric_tensor(QGTOnTheFly)
5  >>> qgt2 = vs.quantum_geometric_tensor(QGTJacobianPyTree(holomorphic=True))
```

---

[15]The number of steps required to find a solution with an iterative linear solver grows with the condition number of the matrix. Therefore, an ill-conditioned matrix requires many steps of iterative solver. For a discussion on this issue, see the paragraph on *Linear Systems* of this section.

| Implementation | Extra arguments | Use-cases | Limitations |
|:---:|:---:|:---|:---|
| `QGTOnTheFly` | `rescale_shift` | shallow networks, large numbers of parameters and samples, few solver steps | |
| `QGTJacobianDense` | `mode` `holomorphic` | deep networks with narrow layers | requires homogeneous parameter types, memory bound |
| `QGTJacobianPyTree` | | deep networks with heterogeneous parameters | memory bound |

Table 3: Overview of the three QGT implementations currently provided by NETKET with their respective options and limitations.

```
6  >>> qgt3 = vs.quantum_geometric_tensor(QGTJacobianDense)
```

**Holomorphicity.**  When performing time evolution or natural gradient descent, one does not always need the full quantum geometric tensor: for ansätze with real parameters, as well as in the case of non-holomorphic wave functions,[16] only the real part of the QGT is used. The real and imaginary parts of the QGT are only required when working with a holomorphic ansatz. (An in-depth discussion of why this is the case can be found at [65, Table 1].)  For this reason, NETKET's QGT implementations return the full geometric tensor only for holomorphic complex-parameter ansätze, and its real part in all other cases.

### 3.5.2   Solving linear systems

For most applications involving the QGT, a linear system of equations of the kind

$$\sum_j G_{ij}\delta_j = f_i \tag{24}$$

needs to be solved, where $G_{i,j}$ is the quantum geometric tensor of a NQS, and $f_i$ is a gradient. This can be done using the standard JAX/NumPy functions, assuming $f$ is a pytree with the same structure as the variational parameters:

```
1  >>> # iterative solver
2  >>> x, info = jax.scipy.sparse.linalg.cg(qgt, f)
3  >>> # direct solver, acting on the dense matrix
4  >>> x, info = jax.numpy.linalg.cholesky(qgt.to_dense(), f)
```

However, we recommend that users call the `solve` method on the QGT object, which allows some additional optimization that may improve performance:

```
1  >>> x, info = qgt.solve(jax.scipy.sparse.linalg.cg, f)
2  >>> x, info = qgt.solve(nk.optimizer.solver.cholesky, f)
```

---

[16]Non-holomorphic functions of complex parameters are internally handled by both JAX and NETKET as real-parameter functions that take the real and imaginary parts of the "complex parameters" separately.

While this works with any of the representations it is advisable to only use Jacobian based implementations ( `QGTJacobianPyTree` or `QGTJacobianDense` ) with direct solvers, since constructing the QGT matrix from `QGTOnTheFly` requires multiplication with all basis vectors, which is not as efficient. Finally, we highlight the fact that users can write their own functions to solve the linear system (24) using advanced regularization schemes (see for instance ref. [22]) and use them together with `qgt.solve` , as long as they respect the standard NumPy solver interface.

When working with iterative solvers such as *cg*, *gmres* or *minres*, the number of steps required to find a solution grows with the condition number of the matrix. Therefore, an ill-conditioned geometric tensor requires many steps of iterative solver, increasing the computational cost. Even then or when using non-iterative methods such as singular value decomposition (SVD), the high condition number can cause instabilities by amplifying noise in the right-hand side of the linear equation [21, 22, 24]. This is especially true for NQS, which typically feature QGTs with a spectrum spanning many orders of magnitude [61], often making QGT-based algorithms challenging to stabilize [16, 22, 24].

To counter that, there is empirical evidence that in some situations, increasing the number of samples used to estimate the QGT and gradients helps to stabilize the solution [24]. Furthermore, it is possible to apply various regularization techniques to the equation. A standard option is to add a small diagonal shift $\epsilon$ to the QGT matrix before inverting it, thus solving the linear equation

$$\sum_j (G_{i,j} + \epsilon)\delta'_j = f_i. \tag{25}$$

When $\epsilon$ is small, the solution $\delta'$ will be close to the desired solution. Otherwise it is biased towards the plain force $f$, which is still acceptable in gradient-based optimization. To add this diagonal shift in NETKET, one of the following approaches can be used:

```
1  >>> qgt_1 = vs.quantum_geometric_tensor(QGTOnTheFly(diag_shift=0.001))
2  QGTOnTheFly(diag_shift=0.001)
3  >>> qgt_2 = qgt_1.replace(diag_shift=0.005)
4  QGTOnTheFly(diag_shift=0.005)
5  >>> qgt_3 = qgt_2 + 0.005
6  QGTOnTheFly(diag_shift=0.01)
```

Regularizing the QGT with a diagonal shift is an effective technique that can be used when performing SR/natural gradient descent for ground state search (see Section 4.1). Note, however, that since the diagonal shift biases the solution of the linear equation towards the plain gradient, it may bias the evolution of the system away from the physical trajectory in cases such as real-time evolution. In those cases, non-iterative solvers such as those based on SVD can be used, the stability of which can be controlled by suppressing smaller singular values. It has also been suggested in the literature to improve stability by suppressing particularly noisy gradient components [22, 40]. This is not currently implemented in NetKet, but planned for a future release. SVD-based regularization also comes at the cost of potentially suppressing physically relevant dynamics [24], making it necessary to find the right balance between stabilization and physical accuracy, and increased computational time as SVD is usually less efficient than iterative solvers.

# 4 Algorithms for variational states

The main use case of NETKET is variational optimization of wave function ansätze. In the current version NETKET, three algorithms are provided out of the box via high-level

driver classes: variational Monte Carlo (VMC) for finding ground states of (Hermitian) Hamiltonians, time-dependent variational Monte Carlo (t-VMC) for real- and imaginary-time evolution, and steady-state search of Liouvillian open-system dynamics.

These drivers are part of the `nk.driver` module but we also export them from the `nk` namespace. They are constructed from the relevant physical model (e.g., a Hamiltonian), a variational state, and other objects used by the optimization method. They all support the `run` method, which performs a number of optimization steps and logs their progress (e.g., variational energies and network parameters) in a variety of output formats.

We highlight that these drivers are built on top of the functionalities described in Sections 2 and 3, and users are free to implement their own drivers or optimization loops, as demonstrated in Section 4.4.

## 4.1 Ground-state search

NETKET provides the variational driver `nk.VMC` for searching for minimal-energy states using VMC [53]. In the simplest case, the `VMC` constructor takes three arguments: the Hamiltonian, an optimizer and the variational state (see Section 3.1). NETKET makes use of optimizers provided by the JAX-based OPTAX library [66],[17] which can be directly passed to `VMC`, allowing the user to build complex training schedules or custom optimizers. In each optimization step, new samples of the variational state are drawn and used to estimate the gradient of the Hamiltonian with respect to the parameters $\theta$ of the ansatz [53] based on the force vector [compare Eq. (14)]

$$f_i = \text{Cov}[O_i, \tilde{H}] = \mathbb{E}[O_i^*(\tilde{H} - \mathbb{E}[\tilde{H}])], \tag{26}$$

where $\tilde{H}$ is the local estimator (12) of the Hamiltonian, known as the *local energy*, and $O_i$ is the log-derivative (13) of the wave function. All expectation values in Eq. (26) are evaluated over the Born distribution $\propto |\psi(\cdot)|^2$ and can therefore be estimated by averaging over the Monte Carlo samples. Given the vector $f$, the direction of steepest descent is given by the energy gradient

$$\nabla_\theta \langle \hat{H} \rangle = 2 \, \text{Re}[f] \quad \text{(real)} \tag{27}$$

or complex co-gradient [44]

$$\nabla_{\theta^*} \langle \hat{H} \rangle = f \quad \text{(complex holomorphic)}. \tag{28}$$

Here we have distinguished the case of i) real parameters and ii) complex parameters with a variational mapping that is holomorphic with respect to $\theta$. For non-holomorphic ansätze (cf. Section 3.2.2), complex parameters can be treated pairs of separate real-valued parameters (real and imaginary part) in the sense of eq. (27). Therefore, this case can be considered equivalent to the real parameter case.

The gradients are then passed on to the OPTAX optimizer, which may transform them (using, e.g., Adam) further before updating the parameters $\theta$. Using the simple stochastic gradient descent optimizer `optax.sgd` (alias `nk.optimizer.Sgd`), the update rule is

$$\theta_i \mapsto \theta_i - \eta f_i. \tag{29}$$

Below we give a short snippet showing how to use the VMC `driver` to find the ground-state of the Ising Hamiltonian.

---

[17]The `nk.optimizer` submodule includes a few optimizers for ease of use and backward compatibility: these are simply re-exports from OPTAX.

```
1   # Define the geometry of the lattice
2   g = nk.graph.Hypercube(length=10, n_dim=1, pbc=False)
3   # Hilbert space of spins on the graph
4   hi = nk.hilbert.Spin(s=1 / 2, N=g.n_nodes)
5   # Construct the Hamiltonian
6   hamiltonian = nk.operator.Ising(hi, graph=g, h=0.5)
7
8   # define a variational state with a Metropolis Sampler
9   sa = nk.sampler.MetropolisLocal(hi)
10  vstate = nk.vqs.MCState(sa, nk.models.RBM())
11
12  # Construct the VMC driver
13  vmc = nk.VMC(hamiltonian,
14              nk.optimizer.Sgd(learning_rate=0.1),
15              variational_state=vstate)
16
17  # run the optimisation for 300 steps
18  output = vmc.run(300)
```

To improve on plain stochastic gradient descent, the `VMC` interface allows passing a keyword argument `preconditioner`. This must be a function that maps a variational state and the gradient vector $f_i$ to the vector $\delta_i$ to be passed to the optimizer as gradients instead of $f_i$. An important use case is *stochastic reconfiguration* [53], where the gradient is preconditioned by solving the linear system of equations

$$\sum_j \mathrm{Re}[G_{ij}]\delta_j = \mathrm{Re}[f_i] \quad \text{(real)} \tag{30}$$

or

$$\sum_j G_{ij}\delta_j = f_i \quad \text{(complex holomorphic)}. \tag{31}$$

The corresponding preconditioner can be created from a QGT class and a JAX-compatible linear solver (the default is `jax.scipy.sparse.linalg.cg`) using `nk.optimizer.SR`:

```
1   # Construct the SR object with the chosen algorithm
2   sr = nk.optimizer.SR(
3   qgt = nk.optimizer.qgt.QGTOnTheFly,
4       solver=jax.scipy.sparse.linalg.bicgstab,
5       diag_shift=0.01,
6   )
7
8   # Construct the VMC driver
9   vmc = nk.VMC(
10      hamiltonian,                            # The Hamiltonian to optimize
11      nk.optimizer.sgd(learning_rate=0.1),    # The optimizer
12      variational_state=vstate,               # The variational state
13      preconditioner=sr,                      # The preconditioner
14  )
```

## 4.2 Finding steady states

In order to study open quantum systems, NETKET provides the `nk.SteadyState` variational driver for determining the variational steady-state $\hat{\rho}_{ss}$ defined as the stationary

point of an arbitrary super-operator $\mathcal{L}$,

$$0 = \frac{d\hat{\rho}}{dt} = \mathcal{L}\hat{\rho}. \tag{32}$$

The search is performed by minimizing the Frobenius norm of the time-derivative [25], which defines the cost function

$$\mathcal{C}(\theta) = \frac{\|\mathcal{L}\hat{\rho}\|_2^2}{\|\hat{\rho}\|_2^2} = \frac{\operatorname{Tr}\left[\hat{\rho}^\dagger \mathcal{L}^\dagger \mathcal{L}\hat{\rho}\right]}{\operatorname{Tr}\left[\hat{\rho}^\dagger \hat{\rho}\right]}, \tag{33}$$

which has a global minimum for the steady state. The stochastic gradient is estimated over the probability distribution of the entries of the vectorized density matrix according to the formula:

$$f_i \equiv \frac{\partial}{\partial \theta_i^*} \frac{\|\mathcal{L}\hat{\rho}\|_2^2}{\|\hat{\rho}\|_2^2} = \mathbb{E}\left[\tilde{\mathcal{L}}\nabla_i^*\tilde{\mathcal{L}}\right] - \mathbb{E}[O_i^*\tilde{\mathcal{L}}^2], \tag{34}$$

where $\tilde{\mathcal{L}}(s, s') = \sum_{m,m'} \mathcal{L}(s, s'; m, m')\rho(m, m')/\rho(s, s')$ is the local estimator proposed in [25], and the expectation values are taken with respect to the "Born distribution" of the vectorized density matrix, $p(s, s') \propto |\rho(s, s')|^2$. The optimization works like the ground-state optimization provided by `nk.VMC` : the gradient is passed to an OPTAX optimizer, which may transform it further before updating the parameters $\theta$. The simplest optimizer, `optax.sgd` , would update the parameters according to the equation

$$\theta_i \to \theta_i - \eta f_i. \tag{35}$$

To improve the performance of the optimization, it is possible to pass the keyword argument `preconditioner` to specify a gradient preconditioner, such as *stochastic reconfiguration* that uses the quantum geometric tensor to transform the gradient. The geometric tensor is computed according to the $L_2$ norm of the vectorized density matrix (see Section 3.5).

As an example, we provide a snippet to study the steady state of a transverse-field Ising chain with 10 spins and spin relaxation corresponding to the Lindblad master equation

$$\mathcal{L}\hat{\rho} = -i\left[\hat{H}, \hat{\rho}\right] + \sum_i \hat{\sigma}_i^- \hat{\rho}\hat{\sigma}_i^+ - \frac{1}{2}\left\{\hat{\sigma}_i^+ \hat{\sigma}_i^-, \hat{\rho}\right\}. \tag{36}$$

We first define the Hamiltonian and a list of jump operators, which are stored in a `LocalLiouvillian` object, which is a lazy representation of the super-operator $\mathcal{L}$. Next, a variational mixed state is built by defining a sampler over the doubled Hilbert space and optionally a different sampler for the diagonal distribution $p(s) \propto \rho(s, s)$, which is used to estimate expectation values of operators. The number of samples used to estimate super-operators and operators can be specified separately, as shown in the example by specifying `n_samples` and `n_samples_diag` .

```
1   # Define the geometry of the lattice
2   g = nk.graph.Hypercube(length=10, n_dim=1, pbc=False)
3   # Hilbert space of spins on the graph
4   hi = nk.hilbert.Spin(s=1 / 2, N=g.n_nodes)
5
6   # Construct the Liouvillian Master Equation
7   ha = nk.operator.Ising(hi, graph=g, h=0.5)
8   j_ops = [nk.operator.spin.sigmam(hi, i) for i in range(g.n_nodes)]
9   # Create the Liouvillian with Hamiltonian and jump operators
10  lind = nk.operator.LocalLiouvillian(ha, j_ops)
```

```
11
12  # Observable
13  sz = sum([nk.operator.spin.sigmam(hi, i) for i in range(g.n_nodes)])
14
15  # Neural Density Matrix
16  sa = nk.sampler.MetropolisLocal(lind.hilbert)
17  vs = nk.vqs.MCMixedState(
18      sa, nk.models.NDM(beta=1), n_samples=2000, n_samples_diag=500
19  )
20  # Optimizer
21  op = nk.optimizer.Sgd(0.01)
22  sr = nk.optimizer.SR(diag_shift=0.01)
23
24  ss = nk.SteadyState(lind, op, variational_state=vs, preconditioner=sr)
25  out = ss.run(n_iter=300, obs={"Sz": sz})
```

### 4.3   Time propagation

Time propagation of variational states can be performed by incorporating the time dependence in the variational parameters and deriving an equation of motion that gives a trajectory in parameters space $\theta(t)$ that approximates the desired quantum dynamics. For real-time dynamics of pure and mixed NQS, such an equation of motion can be derived from the time-dependent variational principle (TDVP) [65, 67, 68]. When combined with VMC sampling to estimate the equation of motion (EOM), this is known as time-dependent variational Monte Carlo (t-VMC) [53, 69] and is the primary approach currently used in NQS literature [8, 19, 20, 22–24]. For complex holomorphic parametrizations[18], the TDVP equation of motion is

$$\sum_j G_{ij}(\theta)\,\dot{\theta}_j = \gamma f_i(\theta, t), \tag{37}$$

with the QGT $G$ and force vector $f$ defined in Sections 4.1 and 4.2. After solving Eq. (37), the resulting parameter derivative $\dot{\theta}$ can be passed to an ODE solver. The factor $\gamma$ determines the type of evolution:

- For $\gamma = -i$, the EOM approximates the real-time Schrödinger equation on the variational manifold, the simulation of which is the main use case for the t-VMC implementation provided by NETKET.

- For $\gamma = -1$, the EOM approximates the imaginary-time Schrödinger equation on the variational manifold. When solved using the first-order Euler scheme $\theta(t + dt) = \theta(t) + \dot{\theta}\,dt$, this EOM is equivalent to stochastic reconfiguration with learning rate $dt$. Imaginary-time propagation with higher-order ODE solvers can therefore also be used for ground state search as an alternative to VMC. This can result in improved convergence in some cases [18].

- For $\gamma = 1$ and with the Lindbladian super-operator taking the place of the Hamiltonian in the definition of the force $f$, this ansatz yields the dissipative real-time evolution according to the Gorini-Kossakowski-Lindblad-Sudarshan master equation [70]. Our implementation uses the QGT induced by the vector norm [26] as discussed in the last paragraph of Section 3.5.

---

[18]The TDVP can be implemented for real-parameter wavefunctions by taking real parts of the right-hand side and QGT similar to VMC (Section 4.1) [40, 47]. This is not yet available in the current version of NETKET, but will be added in a future release.

The current version of NETKET provides a set of Runge–Kutta (RK) solvers based on JAX and a driver class `TDVP` implementing the t-VMC algorithm for the three use cases listed above. At the time of writing, these features are provided as a preview version in the `netket.experimental` namespace as their API is still subject to ongoing development. The ODE solvers are located in the submodule `netket.experimental.dynamics`, the driver under `netket.experimental.TDVP`.

Runge-Kutta solvers implement the propagation scheme [71]

$$\theta(t + dt) = \theta(t) + dt \sum\nolimits_{l=1}^{L} b_l k_l \tag{38}$$

using a linear combination of slopes

$$k_l = F\left(\theta(t) + \sum\nolimits_{m=1}^{l-1} a_{lm} k_m, \ t + c_l \, dt\right), \tag{39}$$

each determined by the solution $F(\theta, t) = \dot{\theta}$ of the equation of motion (37) at an intermediate time step. The coefficients $\{a_{lm}\}$, $\{b_l\}$, and $\{c_l\}$ determine the specific RK scheme and its order. NETKET further supports step size control when using adaptive RK schemes. In this case, the step size is dynamically adjusted based on an embedded error estimate that can be computed with little overhead during the RK step (38) [71]. Step size control requires a norm on the parameters space in order to estimate the magnitude of the error. Typically, the Euclidean norm $\|\delta\| = \sqrt{\sum_i |\delta_i|^2}$ is used. However, since different directions in parameters space influence the quantum state to different degrees, it can be beneficial to use the norm $\|\delta\|_G = \sqrt{\sum_i \delta_i^* G_{ij} \delta_j}$ induced by the QGT $G$ as suggested in Ref. [22], which takes this curvature into account and is also provided as an option with the NETKET time-evolution driver.

An example demonstrating the use of NETKET's time evolution functionality is provided in Sec. 9.

## 4.4  Implementing custom algorithms using NetKet

While key algorithms for energy optimization, steady states, and time propagation are provided out of the box in the current NETKET version, there are many more applications of NQS. While we wish to provide new high-level driver classes for additional use cases, such as quantum state tomography [32] or general overlap optimization [33], it is already possible and encouraged for users to implement their own algorithms on top of NETKET. For this reason, we provide the core building blocks of NQS algorithms in a composable fashion.

For example, it is possible to write a simple loop that solves the TDVP equation of motion (37) for a holomorphic variational ansatz and using a first-order Euler scheme [i.e., $\theta(t + dt) = \theta(t) + \dot{\theta}(t) \, dt$] very concisely, making use of the elementary building blocks provided by the `VariationalState` class:

```
def custom_simple_tdvp(
    hamiltonian: AbstractOperator,  # Hamiltonian
    vstate: VariationalState,       # variational state
    t0: float,                      # initial time
    dt: float,                      # time step
    t_end: float,                   # end time
):
    t = t0
    while t < t_end:
        # compute the energy gradient f
```

```
11          energy, f = vstate.expect_and_grad(hamiltonian)
12          G = vstate.quantum_geometric_tensor()
13          # multiply the gradient by -1.0j for unitary dynamics
14          gamma_f = jax.tree_map(lambda x: -1.0j * x, f)
15          # Solve the linear system using any solver, such as CG
16          # (or write your own regularization scheme)
17          dtheta, _ = G.solve(jax.scipy.sparse.linalg.cg, gamma_f)
18          # update the parameters (theta = theta + dt * dtheta)
19          vs.parameters = jax.tree_map(
20              lambda x, y: x + dt * y, vs.parameters, dtheta
21          )
22          t = t + dt
```

While the included `TDVP` driver (Section 4.3) provides many additional features (such as error handling, step size control, or higher-order integrators) and makes use of JAX's just-in-time compilation, this simple implementation already provides basic functionality and shows how NETKET can be used for quick prototyping.

## 5   Symmetry-aware neural quantum states

NETKET includes a powerful set of utilities for implementing NQS ansätze that are symmetric or transform correctly under the action of certain discrete symmetry groups. Only groups that are isomorphic to a set of permutations of the computational basis are supported. This is useful for modeling symmetric (e.g., lattice) Hamiltonians, whose eigenstates transform under irreducible representations of their symmetry groups. Restricting the Hilbert space to individual symmetry sectors can improve the convergence of variational optimization [72] and the accuracy of its result [15, 17, 50, 73]. Additionally, symmetry restrictions can be used to find excited states [14, 17, 31], provided they are the lowest energy level in a particular symmetry sector.

While there is a growing interest for other symmetry groups, such as continuous ones like $SU(2)$ or $SO(3)$, they cannot be compactly represented in the computational basis and therefore the approach described in this chapter cannot be used. Finding efficient encodings for continuous groups is still an open research problem and it's not yet clear which strategy will work best [17].

NETKET uses *group convolutional neural networks* (GCNNs) to build wave functions that are symmetric over a finite group $G$. GCNNs generalize convolutional neural networks, invariant under the Abelian translation group, to general symmetry groups $G$ which may contain non-commuting elements [74]. GCNNs were originally designed to be invariant, but they can be modified to transform under an arbitrary irreducible representation (irrep) of $G$, using the projection operator [75]

$$|\psi_\chi\rangle = \frac{d_\chi}{|G|} \sum_{g \in G} \chi_g^* \, g|\psi\rangle, \tag{40}$$

where $g$ runs over all symmetry operations in $G$, with corresponding characters $\chi_g$. Under the trivial irrep, where all characters are unity, the invariant model is recovered.

NETKET can infer the full space group of a lattice, defined as a set of permutations of lattice sites, starting from a geometric description of its point group. It can also generate nontrivial irrep characters [to be used in (40) for states with nonzero wave vectors or transforming nontrivially under point-group symmetries] using a convenient interface that approximates standard crystallographic formalism [76]. In addition, NETKET pro-

vides powerful group-theoretic algorithms for arbitrary permutation groups of lattice sites, allowing new symmetry elements to be easily defined.

Pre-built GCNNs are then provided in the `nk.models` submodule, which can be constructed by specifying few parameters, such as the number of features in each layer, and the lattice or permutation group under which the network should transform. Symmetric RBMs [8] are also implemented as one-layer GCNNs that aggregate convolutional features using a product rather than a sum. These pre-built network architectures are made up of individual layers found in the `nk.nn` submodule, which can be used directly to build custom symmetric ansätze.

Section 5.1 describes the NETKET interface for constructing space groups of lattices and their irreps. Usage of GCNNs is described in Section 5.2, while appendix A provides mathematical and implementation details.

## 5.1   Symmetry groups and representation theory

NETKET supports symmetry groups that act on a discrete Hilbert space defined on a lattice. On such a Hilbert space, space-group symmetries act by permuting sites; most generally, therefore, arbitrary subgroups of the symmetric group $S_N$ of a lattice of $N$ sites are supported. A symmetry group can be specified directly as a list of permutations, as in the following example, which enforces the symmetry $\psi(s_0, s_1, s_2, s_3) = \psi(s_3, s_1, s_2, s_0)$ for all four-spin configurations $s = (s_0, \ldots, s_3)$, $s_i = \pm 1$:

```
1  hi = nk.hilbert.Spin(1/2, N=4)
2  symms = [
3      [0, 1, 2, 3],   # identity element
4      [3, 1, 2, 0],   # swap first and last site
5  ]
6  model = nk.models.RBMSymm(symms, alpha=1)
```

The listed permutations are required to form a group and, in particular, the identity operation $e : s \mapsto s$ must always be included as the first element.

It is inconvenient and error-prone to specify all space-group symmetries of a large lattice by their indices. Therefore, NETKET provides support for abstract representations of permutation and point groups through the `nk.utils.group` module, complete with algorithms to compute irreducible representations [77–79]. The module also contains a library of two- and three-dimensional point groups, which can be turned into lattice-site permutation groups using the graph class `nk.graph.Lattice` (but not general `Graph` objects, for they carry no geometric information about the system):

```
1  from netket.utils.group.planar import D
2  from netket.graph import Lattice
3
4  # construct a centred rectangular lattice
5  lattice = Lattice(
6      basis_vectors = [[2,0], [0,1]], # each row is a lattice vector
7      extent = (5,5),
8      site_offsets = [[0,0], [1,0.5]], # each row is the position of a site in
         the unit cell
9      point_group = D(2) # the point group of the lattice, here Z_2 x Z_2
10 )
```

NETKET contains specialized constructors for some lattices (e.g., `Square` or `Pyrochlore`), which come with a default point group; however, these can be overridden in methods like `Lattice.space_group`:

```
1  from netket.utils.group.planar import rotation, reflection_group, D
2  from netket.utils.group import PointGroup, Identity
3  from netket.graph import Honeycomb
4
5  # construct the D_6 point group of the honeycomb lattice by hand
6  cyclic_6 = PointGroup(
7      [Identity()] + [rotation(360 / n * i) for i in range(1, n)],
8      ndim=2,
9  )
10 # the @ operator returns the Cartesian product of groups
11 # but doesn't check for group structure
12 dihedral_6 = reflection_group(angle=0) @ cyclic_6
13
14 assert dihedral_6 == D(6)
15
16 lattice = Honeycomb([6,6])
17
18 # returns the full space group of `lattice` as a PermutationGroup
19 space_group = lattice.space_group()
20 # the space group is spanned by 6^2 translations and 12 point-group
       symmetries
21 assert len(space_group) == 12 * 6 * 6
22
23 # do this if the Hamiltonian breaks reflection symmetry
24 # can also be used for generic Lattices that have no default point group
25 space_group = lattice.space_group(cyclic_6)
```

Irreducible representation (irrep) matrices can be computed for any point or permutation group object using the method `irrep_matrices()` . Characters (the traces of these matrices) are returned by the method `character_table()` as a matrix, each row of which lists the characters of all group elements. Character tables closer to the format familiar from quantum chemistry texts are produced by `character_table_readable()` . Irrep matrices and character tables are calculated using adaptations of Dixon's [77] and Burnside's [78] algorithms, respectively.

It would, however, be impractical to inspect irreps of a large space group directly to specify the symmetry sector on which to project a GCNN wave function. Exploiting the semidirect-product structure of space groups [79], space-group irreps are usually[19] described in terms of a set of symmetry-related wave vectors (known as a *star*) and an irrep of the subgroup of the point group that leaves the same invariant (known as the *little group*) [76]. Irreps can be constructed in this paradigm using `SpaceGroupBuilder` objects returned by `Lattice.space_group_builder()` :

```
1  from netket.graph import Triangular
2
3  lattice = Triangular([6,6])
4  momentum = [0,0]
5  # space_group_builder() takes an optional PointGroup argument
6  sgb = lattice.space_group_builder()
7
8  # choosing a representation
9  # this one corresponds to the B_2 irrep at the Gamma point
```

---

[19]Representation theory for wave vectors on the surface of the Brillouin zone in a nonsymmorphic space group is much more complicated [79] and is not currently implemented in NETKET.

```
10  chi = sgb.space_group_irreps(momentum)[3]
```

The irrep `chi`, generated using the little group, is equivalent to one of the irreps in `Lattice.space_group().character_table()` and can thus be used for symmetry-projecting GCNN ansätze. The order in which irreps of the little group are returned can readily be checked in an interactive session:

```
1  >>> sgb.little_group(momentum).character_table_readable()
2  (['1xId()', '2xRot(60)', '2xRot(120)', '1xRot(180)', '3xRefl(0)',
      '3xRefl(-30)'],
3  array([[ 1.,  1.,  1.,  1.,  1.,  1.],      # this is irrep A1
4         [ 1.,  1.,  1.,  1., -1., -1.],      # A2
5         [ 1., -1.,  1., -1.,  1., -1.],      # B1
6         [ 1., -1.,  1., -1., -1.,  1.],      # B2
7         [ 2.,  1., -1., -2.,  0.,  0.],      # E1
8         [ 2., -1., -1.,  2.,  0.,  0.]]))    # E2
```

The main caveat in using this machinery is that the point groups predefined in NETKET all leave the origin invariant (except for `cubic.Fd3m` which represents the "nonsymmorphic point group" of the diamond/pyrochlore lattice) and thus only work well with lattices in which the origin has full point-group symmetry. This behaviour can be changed (see the definition of `cubic.Fd3m` for an example), but it is generally safer to define lattices using the proper Wyckoff positions [76], of which the origin is usually maximally symmetric.

## 5.2   Using group convolutional neural networks (GCNNs)

NETKET uses GCNNs [50, 74] to create NQS ansätze that are symmetric under space groups of lattices. These networks consist of alternating *group convolutional layers* and pointwise nonlinearities. The former can be thought of as a generalization of convolutional layers to a generic finite group $G$. They are *equivariant,* that is, if their inputs are transformed by some space-group symmetry, features in all subsequent layers are transformed accordingly. As a result, the output of a GCNN can be understood as amplitudes of the wave functions $g|\psi\rangle$ for all $g \in G$, which can be combined into a symmetric wave function using the projection operator (40). Further details about equivariance and group convolutions are given in Appendix A.1.

GCNNs are constructed by the function `nk.models.GCNN`. Symmetries are specified either as a `PermutationGroup` or a `Lattice`. In the latter case, the symmetry group is given by `space_group()`; an optional `point_group` argument to `GCNN` can be used to override the default point group. By default, output transforms under the trivial irrep $\chi_g \equiv 1$, that is, all output features are averaged together to obtain a wave function that is fully symmetric under the whole space group. Other irreps can be specified through the `characters` argument, which takes a vector of the same size as the space group.

```
1  from netket.graph import Triangular
2  from netket.models import GCNN
3  from netket.utils.group.planar import C
4
5  lattice = Triangular([6,6])
6  momentum = [0,0]
7  sgb = lattice.space_group_builder()
8  chi = sgb.space_group_irreps(momentum)[3]
9
10 # This transforms as the trivial irrep Gamma A_1
11 gcnn1 = GCNN(lattice, layers = 4, features=4)
```

|  | `mode="irreps"` | `mode="fft"` |
|---|---|---|
| Can be used for | any group | only space groups |
| Symmetries can be specified by | • `Lattice`<br>• `PermutationGroup`<br>• Symmetry permutations and irrep matrices | • `Lattice`<br>• `PermutationGroup` and shape of translation group<br>• Symmetry permutations, product table, and shape of translation group |
| Kernel memory footprint per layer | $O(f_{\text{in}}f_{\text{out}}\|G\|)$ | $O(f_{\text{in}}f_{\text{out}}\|G\|\|P\|)$ |
| Evaluation time per layer per sample | $O[(f_{\text{in}} + f_{\text{out}})\|G\|^2 + f_{\text{in}}f_{\text{out}}\sum_a d_a^3]$ | $O[(f_{\text{in}} + f_{\text{out}})\|G\|\log\|T\| + f_{\text{in}}f_{\text{out}}\|G\|\|P\|]$ |
| Preferable for | • large point groups<br>• if expanded `"fft"` kernels don't fit in memory | • small point groups<br>• very large batches (see App. A.2) |

Table 4: Comparison of GCNN implementations. $f_{\text{in,out}}$ stands for the number of input and output features, $\|G\|, \|P\|, \|T\|$ for the sizes of the space group, point group, and translation group, respectively. $d_a$ are the dimensions of irreps of $G$; in a large space group, $\sum_a d_a^3$ scales as $\|G\|\|P\|$.

```
12
13  # This transforms as Gamma B_2
14  gcnn2 = GCNN(lattice.space_group(), characters=chi, layers=4, features=4)
15
16  # This does not enforce reflection symmetry
17  gcnn3 = GCNN(lattice, point_group=C(6), layers=4, features=4)
```

NETKET currently supports two implementations of GCNNs, one based on group Fourier transforms ( `mode="irreps"` ), the other using fast Fourier transforms on each coset of the translation group ( `mode="fft"` ): these are discussed in more detail in Appendix A.2. Their behavior is equivalent, but their performance and calling sequence is different, as explained in Table 4. A default `mode="auto"` is also available. For spin models, parity symmetry (taking $\sigma^z$ to $-\sigma^z$) is a useful extension of the U(1) spin symmetry group enforced by fixing magnetization along the $\sigma^z$ axis. Parity-enforcing GCNNs can be constructed using the `parity` argument, which can be set to $\pm 1$.

In addition to deep GCNNs, fully symmetric RBMs [8] are implemented in `nk.models.RBMSymm` as a single-layer GCNN from which the wave function is computed as

$$\psi = \prod_{i,g\in G} \cosh f_g^{(i)} \implies \log\psi = \sum_{i,g\in G} \ln\cosh f_g^{(i)}. \tag{41}$$

Due to the products (rather than sums) used, this ansatz only supports wave functions that transform under the trivial irrep. An RBM-like structure closer to that of ref. [73] can be achieved using a single-layer GCNN:

```
1  from netket.models import GCNN, RBMSymm
2  from netket.nn import logcosh
3
```

```
4   # fully symmetric RBM
5   rbm1 = RBMSymm(group, alpha=4)
6
7   # symmetrized RBM similar to (Nomura, 2021)
8   rbm2 = GCNN(group, layers=1, features=4, output_activation=logcosh)
```

# 6 Quantum systems with continuous degrees of freedom

In this section we will introduce the tools provided by NETKET to study systems with continuous degrees of freedom. The interface is very similar to the one introduced in Section 2 for discrete degrees of freedom.

## 6.1 Continuous Hilbert spaces

Similar to the discrete Hilbert spaces, the bosonic Hilbert space of $N$ particles in continuous space has the structure

$$\mathcal{H}_{\text{continuous}} = \text{span}\{|x_0\rangle \otimes \cdots \otimes |x_{N-1}\rangle : x_i \in \mathcal{L}_i, i \in \{0, \ldots, N-1\}\} \tag{42}$$

where $\mathcal{L}_i$ is the space available to each individual boson: for example, $\mathcal{L}_i$ is $\mathbb{R}^d$ for a free particle in $d$ spatial dimensions, and $[0, L]^d$ for particles confined to a $d$-dimensional box of side length $L$. In the case of finite simulation cells, the boundaries can be equipped with periodic boundary conditions.

In the following snippet, we define the Hilbert space of five bosons in two spatial dimensions, confined to $[0, 10]^2$ with periodic boundary conditions:

```
1   >>> hilb = nk.hilbert.Particle(N=5, L=(10.0, 10.0), pbc=True)
2   >>> print("Size of the basis: ", hilb.size)
3   Size of the basis:  10
4   >>> hilb.random_state(nk.jax.PRNGKey(0), (2,))
5   [[0.02952452 0.21660899 2.836163   3.5628846  4.5622005  5.9473248
6     6.104126   8.14864    9.163713   9.263418  ]
7    [9.85617    0.4667458  2.211265   4.1587596  4.250165   6.69916
8     6.5165453  7.3764215  8.508119   0.08060825]]
```

As we discussed in Section 2.1, the Hilbert objects only define the computational basis. For that reason, the flag `pbc=True` only affects what configurations can be generated by samplers and how to compute the distance between two different sets of positions. This option does not enforce any boundary condition for the wave-function, which would have to be accounted for into the variational ansatz.

## 6.2 Linear operators

Similar to the discrete-variable case, expectation values of operators can be estimated as classical averages of the local estimator

$$\tilde{O}(x) = \frac{\langle x| \hat{O} |\psi\rangle}{\langle x|\psi\rangle} \tag{43}$$

over the (continuous) Born distribution $p(x) \propto |\psi(x)|^2$. NETKET provides the base class `ContinuousOperator` to write custom (local) operators and readily implements Hamilto-

nians of the form ($\hbar = 1$ in our units)

$$\hat{H} = -\frac{1}{2} \sum_i \frac{1}{m_i} \nabla_i^2 + \hat{V}(\{\mathbf{x}_i\}) \tag{44}$$

using the predefined operators `KineticEnergy` and `PotentialEnergy`. For example, a harmonically confined system described by $\hat{H} = -\frac{1}{2}\sum_i \nabla_i^2 + \frac{1}{2}\sum_i \hat{\mathbf{x}}_i^2$ can be implemented as

```
1   # This function takes a single vector and returns a scalar
2   def v(x):
3       return 0.5*jnp.linalg.norm(x) ** 2
4
5   # Construct the Kinetic energy term with unit mass
6   H_kin = nk.operator.KineticEnergy(hilb, mass=1.0)
7   # Construct the Potential energy term using the potential defined above
8   H_pot = nk.operator.PotentialEnergy(hilb, v)
9
10  # Sum the two objects into a single Operator
11  H = H_kin + H_pot
```

Operators defined on continuous Hilbert spaces cannot be converted to a matrix form or used in exact diagonalization, in contrast to those defined on discrete Hilbert spaces. Continuous operators can still be used to compute expectation values and their gradients with a variational state.

## 6.3 Samplers

Out of the built-in samplers in the current version of NETKET (Section 3.4), only the Markov chain Monte Carlo sampler `MetropolisSampler` supports continuous degrees of freedom, as both `ExactSampler` and the autoregressive `ARNNSampler` rely on the sampled basis being countable. For continuous spaces, we provide the transition rule `sampler.rules.GaussianRule` which proposes new states by adding a random shift to every degree of freedom sampled from a Gaussian of customizable width. More complex transition rules can be defined following the instructions provided in Section 3.4.

## 6.4 Harmonic oscillators

As a complete example of how to use continuous-space Hilbert spaces, operators, variational states, and the VMC driver together, consider 10 particles in three-dimensional space, confined by a harmonic potential $V(x) = x^2/2$. The exact ground-state energy of this system is known to be $E_0 = 15$. We use the multivariate Gaussian ansatz $\log \psi(x) = -x^{\mathsf{T}} \Sigma^{-1} x$, where $\Sigma = TT^{\mathsf{T}}$ and $T$ is randomly initialized using a Gaussian with zero mean and variance one. Note that the form of $\Sigma$ ensures that it is positive definite.

```
1   import netket as nk
2   import jax.numpy as jnp
3
4   def v(x):
5       return 0.5*jnp.linalg.norm(x) ** 2
6
7   hilb = nk.hilbert.Particle(N=10, L=(jnp.inf,jnp.inf,jnp.inf), pbc=False)
8   ekin = nk.operator.KineticEnergy(hilb, mass=1.0)
```

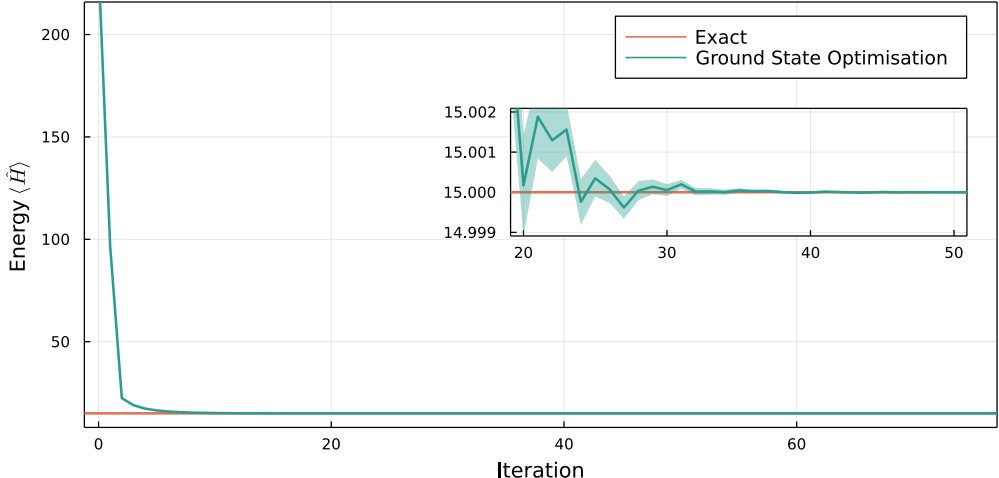

Figure 1: VMC energy estimate as a function of the optimization step for a continuous-space system of $N = 10$ particles in $d = 3$ spatial dimensions subject to a harmonic confinement.

```
9   pot = nk.operator.PotentialEnergy(hilb, v)
10  ha = ekin + pot
11
12  sa = nk.sampler.MetropolisGaussian(hilb, sigma=0.1, n_chains=16, n_sweeps=32)
13  model = nk.models.Gaussian(param_dtype=float)
14  vs = nk.vqs.MCState(sa, model, n_samples=10 ** 4, n_discard=2000)
15
16  op = nk.optimizer.Sgd(0.05)
17  sr = nk.optimizer.SR(diag_shift=0.01)
18
19  gs = nk.VMC(ha, op, sa, variational_state=vs, preconditioner=sr)
20  gs.run(n_iter=100, out="HO_10_3d")
```

We show the training curve of above snippet in Fig. 1; exact ground-state energy is recovered to a very high accuracy.

## 6.5   Interacting system with continuous degrees of freedom

In this example we want to tackle an interacting system of bosonic Helium particles in one continuous spatial dimension. The two-body interaction is given by the *Aziz* potential which qualitatively resembles a Lennard-Jones potential [80–82]. The Hamiltonian reads

$$H = -\frac{\hbar^2}{2m} \sum_i \nabla_i^2 + \sum_{i<j} V_{\text{Aziz}}(r_{ij}) \tag{45}$$

We will examine the system at a density $\rho = \frac{N}{L} = 0.3\text{Å}^{-1}$ with $N = 10$ particles in units where $\hbar = m = k_b = 1$. To confine the system it is placed in a box of size $L$ equipped with periodic boundary conditions. The Hilbert space and sampler are initialized as shown above ($r_m$ is the length-scaled defined in the Aziz potential):

```
1   import netket as nk
2
3   N = 10
4   d = 0.3 # 1/Angstrom
```

```
5   rm = 2.9673 # Angstrom
6   L = N / (0.3 * rm)
7   hilb = nk.hilbert.Particle(N=N, L=L, pbc=True)
8   sab = nk.sampler.MetropolisGaussian(hilb, sigma=0.05, n_chains=16,
        n_sweeps=32)
```

### 6.5.1   Defining the Hamiltonian

We can define the Hamiltonian through the action of the interaction-potential on a sample of positions $x$, and combine it with the predefined kinetic energy operator. Since we are using periodic boundary conditions, we will use the Minimum Image Convention (MIC) to compute distances between particles. In the following snippet the Aziz potential (in the units above) is defined and the Hamiltonian is instantiated:

```
1
2   def minimum_distance(x, sdim):
3       """Computes distances between particles using mimimum image convention"""
4       n_particles = x.shape[0] // sdim
5       x = x.reshape(-1, sdim)
6
7       distances = (-x[jnp.newaxis, :, :] + x[:, jnp.newaxis, :])[
8           jnp.triu_indices(n_particles, 1)
9       ]
10      distances = jnp.remainder(distances + L / 2.0, L) - L / 2.0
11
12      return jnp.linalg.norm(distances, axis=1)
13
14  def potential(x, sdim):
15      """Compute Aziz potential for single sample x"""
16      eps = 7.846373
17      A = 0.544850 * 10 ** 6
18      alpha = 13.353384
19      c6 = 1.37332412
20      c8 = 0.4253785
21      c10 = 0.178100
22      D = 1.241314
23
24      dis = minimum_distance(x, sdim)
25      return jnp.sum(
26          eps
27          * (
28              A * jnp.exp(-alpha * dis)
29              - (c6 / dis ** 6 + c8 / dis ** 8 + c10 / dis ** 10)
30              * jnp.where(dis < D, jnp.exp(-((D / dis - 1) ** 2)), 1.0)
31          )
32      )
33
34  ekin = nk.operator.KineticEnergy(hilb, mass=1.0)
35  pot = nk.operator.PotentialEnergy(hilb, lambda x: potential(x, 1))
36  ha = ekin + pot
```

### 6.5.2   Defining and training the variational Ansatz

There are two properties that the variational Ansatz for this system must obey:

1. It must be invariant with respect to the permutations of its particles, because they are bosons;

2. As the interaction resembles a Lennard-Jones potential we have a strong divergence in the potential energy when particles get close to each other. This divergence must be compensated by the kinetic energy.

We satisfy the permutation-invariance by using a neural network architecture called *DeepSets*. *DeepSets* exploit the fact that any function $f(x_1, ..., x_N)$ which is invariant under permutations of its inputs can be decomposed as [83]:

$$f(x_1, ..., x_N) = \rho\left(\sum_i \phi(x_i)\right) \tag{46}$$

where $\rho$ and $\phi$ are arbitrary functions. In this specific example, $x_i$ denotes a single-particle position and $\rho$ and $\phi$ are parameterized with dense feed forward neural networks.

The second requirement is fulfilled by using *Kato's cusp condition* which states that [84]

$$\lim_{r \to 0}\left(\frac{\nabla^2 \psi_c(r)}{\psi_c(r)} + V(r)\right) < \infty \tag{47}$$

where $r$ denotes the distance between the particles and $\psi_c$ is the cusp wave-function. For the case of a Lennard-Jones potential ($\propto r^{-12}$-divergence), we have

$$\psi_c(r) = \exp\left[-\frac{1}{2}\left(\frac{b}{r}\right)^5\right], \tag{48}$$

where $b$ is a variational parameter.

We also need to handle the periodic conditions, making sure that the wave-function does not exhibit divergent behaviour at the edges of the (periodic) box. To this end we will use a surrogate distance function for the minimum image convention, namely $d_{\sin}(x_i, x_j) = \frac{L}{2}\sin(\frac{\pi}{L}r_{ij})$ in the variational Ansatz. Additionally we replace the single-particle coordinates in Eq. (46) by the two-particle distances $d_{\sin}(x_i, x_j)^2$, such that all in all our variational Ansatz reads

$$\psi(x_1, ..., x_N) = \exp\left[\rho\left(\sum_{i<j}\phi(d_{\sin}(x_i, x_j)^2)\right)\right] \cdot \exp\left[-\frac{1}{2}\left(\frac{b}{d_{\sin}(x_i, x_j)}\right)^5\right] \tag{49}$$

The variational ansatz we described here is implemented in NETKET as `DeepSeetRelDistance`, and a more in-depth discussion can be found in this reference [85].

Having defined the ansatz, we run the VMC driver with the given variatianal Ansatz to find an estimation of the ground-state energy of the system. This is done as follows:

```
1   model = nk.models.DeepSetRelDistance(
2       hilbert=hilb,
3       cusp_exponent=5,
4       layers_phi=2,
5       layers_rho=3,
6       features_phi=(16, 16),
7       features_rho=(16, 16, 1),
8   )
9   vs = nk.vqs.MCState(sab, model, n_samples=4096, n_discard_per_chain=128)
10
11  op = nk.optimizer.Sgd(0.001)
```

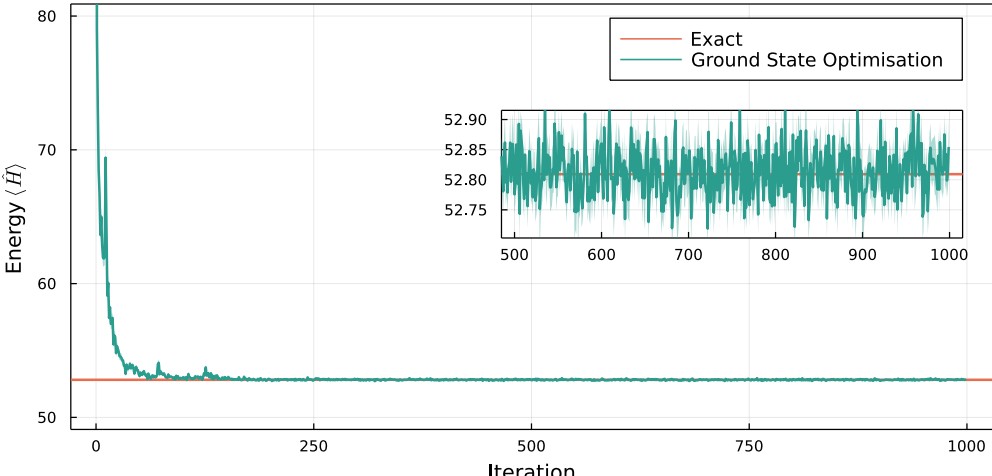

Figure 2: VMC energy estimate as a function of the optimization step for a continuous-space system of $N = 10$ particles in $d = 1$ spatial dimensions subject to a LJ-like interaction potential placed within a periodic box. The green dashed line is the result given in the supplementary material of [81].

```
12  sr = nk.optimizer.SR(diag_shift=0.01)
13
14  gs = nk.VMC(ha, op, sab, variational_state=vs, preconditioner=sr)
15  gs.run(n_iter=1000, out="Helium_10_1d")
```

The result of this optimization and a comparison to literature results is displayed in Fig. 2.

# 7    Example: Finding ground and excited states of a lattice model

In this example, we define the $J_1$–$J_2$ Heisenberg Hamiltonian

$$H = J_1 \sum_{\langle ij \rangle} \vec{\sigma}_i \cdot \vec{\sigma}_j + J_2 \sum_{\langle\langle ij \rangle\rangle} \vec{\sigma}_i \cdot \vec{\sigma}_j, \tag{50}$$

on a $10 \times 10$ square lattice and use the VMC code introduced in Section 4.1 to find a variational approximation of its ground state. This model gives rise to several phases of matter, including magnetically ordered states, a valence bond solid, and a quantum spin liquid. Here, we set $J_1 = 1, J_2 = 0.5$, inside the spin liquid phase [31,86].

Our example is optimized to run on a single GPU with 16 GB of memory. We will make note of what should be changed when running the simulation on CPUs.

## 7.1   Defining the lattice and the Hamiltonian

We use the `Lattice` class to define the square lattice and generate its space-group symmetries. By passing `max_neighbor_order=2` to the constructor, we generate graph edges for both nearest and next-nearest neighbours. The pre-defined `Heisenberg` class supports passing different coupling constants for both types of edge.

```python
import netket as nk
import numpy as np
import json
from math import import pi

L = 10
# Build square lattice with nearest and next-nearest neighbor edges
lattice = nk.graph.Square(L, max_neighbor_order=2)
hi = nk.hilbert.Spin(s=1 / 2, total_sz=0, N=lattice.n_nodes)
# Heisenberg with coupling J=1.0 for nearest neighbors
# and J=0.5 for next-nearest neighbors
H = nk.operator.Heisenberg(hilbert=hi, graph=lattice, J=[1.0, 0.5])
```

## 7.2 Defining and training a symmetric ansatz

To enforce all spatial symmetries of (50), we use the GCNN ansatz described in Section 5. By default, the GCNN projects onto the symmetric representation, which contains the ground state for this geometry. We select singlet states by only sampling basis states with $\sum_i S_i^z = 0$ and setting spin parity using `parity=1`. We use a model with four layers, each containing four feature vectors (i.e., four hidden units for each of the $8L^2$ space-group symmetries). To exploit the high degree of parallelism of GPUs, we set `sampler.n_chains` equal to `vstate.n_samples` [20]. When using CPUs, `n_chains` should be set to a smaller value.

```python
machine = GCNN(
    symmetries=lattice,
    parity=1,
    layers=4,
    features=4,
    param_dtype=np.complex128,
)
sampler = nk.sampler.MetropolisExchange(
    hilbert=hi,
    n_chains=1024,
    graph=lattice,
    d_max=2,
)
opt = nk.optimizer.Sgd(learning_rate=0.02)
sr = nk.optimizer.SR(diag_shift=0.01)
vstate = nk.vqs.MCState(
    sampler=sampler,
    model=machine,
    n_samples=1024,
    n_discard_per_chain=0,
    chunk_size=4096,
)
gs = nk.VMC(H, opt, variational_state=vstate, preconditioner=sr)
gs.run(n_iter=200, out="ground_state")
```

---

[20]Note that this results in a somewhat non-standard MC scheme where, instead of an ensemble of chains with generally non-zero internal autocorrelation, the sampler produces an ensemble of independently drawn single configurations ( `n_samples_per_chain==1` ). Since the sampler state of the previous VMC step is used as initial state for the next step, there can still be a residual autocorrelation with the previous samples, which is, however, alleviated by the sampler performing $N_{\text{sites}}$ intermediate updates before yielding the single requested sample with the settings used here.

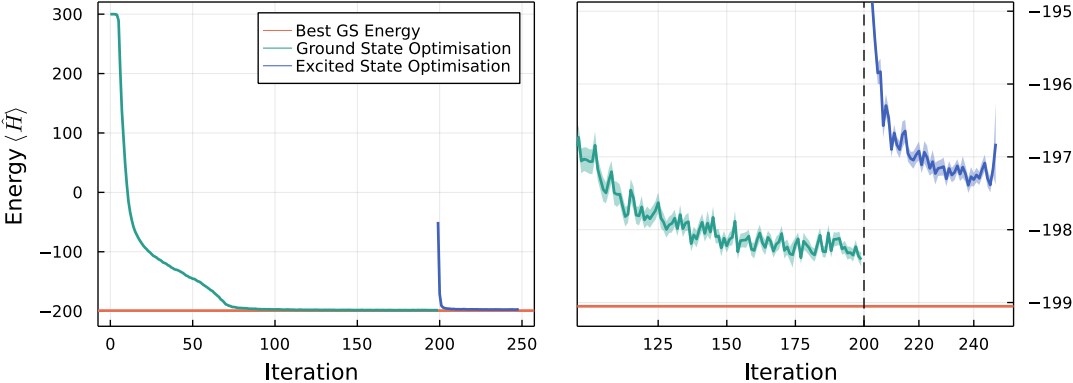

Figure 3: Energy evolution of variational ground states (green) and excited states after transfer learning (blue), compared to the lowest known variational energy for the $10 \times 10$ square-lattice $J_1$–$J_2$ model [31] (black). The variational energies can be further improved by allowing more training steps, Monte Carlo samples, etc. The plot on the right zooms in on the lowest energies.

```
25
26  data = json.load(open("ground_state.log"))
27  print(np.mean(data["Energy"]["Mean"]["real"][-20:])/400)
28  #  Output: -0.49562531096409457
29  print(np.std(data["Energy"]["Mean"]["real"][-20:])/400)
30  #  Output: 0.0002492304032505182
```

We specify `chunk_size=4096` in the variational state in order to reduce memory consumption. As we have $L^2 = 100$ sites, at every VMC step we will need to evaluate the network for $\mathcal{O}(N_{\text{samples}} L^2) = \mathcal{O}(10^3 \cdot 10^2)$ different configurations, but the memory available on commercial GPUs will not be enough to perform this computation in a single pass. Instead, by setting `chunk_size` NETKET will split the calculation in many smaller sub-calculations (see Section 3.3.1 for more details).

This calculation, which takes about 30 minutes on an NVIDIA A100 GPU, already delivers a fairly accurate variational energy. The evolution of variational energy during the training procedure is shown in Fig. 3.

We note that a typical initialization of a GCNN gives rise to ferromagnetic correlations, which can make training an antiferromagnetic Hamiltonian unstable [16, 50]. Therefore, it is often good practice to pre-optimize the phases by restricting all amplitudes to unity by setting `equal_amplitudes=True` switch and training only the phases of the network. These parameters can then be loaded into a model with `equal_amplitudes=False`.

## 7.3 Finding an excited state

We can also find low-lying excited states using this procedure, by projecting the wavefunction onto a different irrep. Here, we consider the first gapless mode in the Anderson tower of states of the Néel antiferromagnet [87], a triplet at wave vector $(\pi, \pi)$ that transforms trivially under all point-group symmetries. This mode is still gapless in the quantum spin liquid; we project out spin-singlets by focusing on parity odd states.

We expedite this calculation by using parameters optimized for the ground-state sector as an initial guess. The resulting wave function will already have a low variational energy (as shown in Fig. 3) and correlations typical for low-energy eigenstates.

```
1   # store the optimized ground-state parameters
2   saved_params = vstate.parameters
3   # Compute the characters of the first excited state
4   characters = lattice.space_group_builder().space_group_irreps(pi, pi)[0]
5   # Construct a model respecting the first-excited state symmetries
6   machine = GCNN(
7       symmetries=lattice,
8       characters=characters,
9       parity=-1,
10      layers=4,
11      features=4,
12      param_dtype=complex,
13  )
14  vstate = nk.vqs.MCState(
15      sampler=sampler,
16      model=machine,
17      n_samples=1024,
18      n_discard_per_chain=0,
19      chunk_size=4096,
20  )
21  # assign the old parameters to the new variational state
22  vstate.parameters = saved_params
23  gs = nk.VMC(H, opt, variational_state=vstate, preconditioner=sr)
24  gs.run(n_iter=50, out='excited_state')
25
26  data = json.load(open("excited_state.log"))
27  print(np.mean(data["Energy"]["Mean"]["real"][-10:])/400)
28  #  Output: -0.49301426054097885
29  print(np.std(data["Energy"]["Mean"]["real"][-10:])/400)
30  #  Output: 0.0003802670752071611
```

# 8   Example: Fermions on a lattice

NETKET can also be used to simulate fermionic systems with a finite number of orbitals. Functionality related to fermions is kept in the `netket.experimental` in order to signal that some parts of the API might still slightly change while we gather feedback from the community. We usually import this namespace as `nkx` as follows:

```
1   from netket import experimental as nkx
```

and then use `nkx` freely.

The Hilbert space for discrete fermionic systems is called `SpinOrbitalFermions`. It supports fermions, with and without a spin- degree of freedom, which occupy a set of orbitals (such as the sites of a lattice). Internally, it uses a tensor product of a `Fock` space for each spin component. For a set of spin-1/2 fermions, we can fix the number of fermions with up (↑) and down (↓) spins through the `n_fermions` keyword argument.

The `SpinOrbitalFermions` generates samples that correspond to occupation numbers $|n\rangle = |n_{1,\uparrow}, ..., n_{N_o,\uparrow}, n_{1,\downarrow}, ..., n_{N_o,\downarrow}\rangle$, for a given ordering of the $N_o$ orbitals (or sites).

In the example below, we determine the ground state of the Fermi-Hubbard model on

a square lattice

$$\hat{H} = -t \sum_{\langle i,j \rangle} \sum_{\sigma=\{\uparrow,\downarrow\}} f_{i,\sigma}^{\dagger} f_{j,\sigma} + h.c. + U \sum_{i} n_{i,\uparrow} n_{i,\downarrow} \tag{51}$$

where $n_{i,\sigma} = f_{i,\sigma}^{\dagger} f_{i,\sigma}$.

NETKET implements a class `FermionOperator2nd` that represents an operator in second quantization. This class does not separate spin and orbital indices. Internally, the `FermionOperator2nd` computes matrix elements of a fermion operator $f_i^{\dagger}$ on an orbital $i$ through the Jordan-Wigner transformation

$$f_i^{\dagger} \rightarrow \left( \bigotimes_{j<i} Z_j \right) \left( \frac{X_i + iY_i}{2} \right) \tag{52}$$

or in terms of occupation numbers

$$\langle n | f_i^{\dagger} | n' \rangle = (-1)^{\sum_{j<i} n_j} \delta_{n_i'+1,n_i} \prod_{j \neq i} \delta_{n_j,n_j'} \tag{53}$$

One can create a `FermionOperator2nd` object from an external `FermionOperator` object from the OPENFERMION library, which is popular for symbolic manipulation of fermionic operators [88]. The small intermezzo code below shows how this works in practice for an operator $f_1^{\dagger} f_2 + 4 f_3 f_0^{\dagger}$

```
from openfermion import FermionOperator
from netket.operator import FermionOperator2nd

of_operator = FermionOperator("1^ 2") + 4*FermionOperator("3 0^")
nk_operator1 = FermionOperator2nd.from_openfermion(of_operator)

nk_operator2 = FermionOperator2nd("1^ 2") + 4*FermionOperator2nd("3 0^")
```

where `nk_operator1` and `nk_operator2` are equivalent.

The mapping between fermions and qubit degrees of freedom is not unique [89], and the Jordan-Wigner transformation is one well-know example of such a transformation. However, by interfacing with toolboxes specialized in symbolic manipulation, we open up a range of possibilities, especially in combination with `PauliStrings.from_openfermion`, which converts `openfermion.QubitOperator` from OPENFERMION to a `PauliStrings` object in NETKET. This allows one to, for example, use a wider range of alternatives to the Jordan-Wigner transformation implemented in OPENFERMION, or other variations.

Going back to our example of the Fermi-Hubbard model, NETKET also implements a more easy to use set of creation and annihilation operators that clearly separate the orbitals and spin indices

- $f_{i,\sigma}^{\dagger}$:  `nkx.operator.fermion.create`

- $f_{i,\sigma}$:  `nkx.operator.fermion.destroy`

- $n_{i,\sigma}$:  `nkx.operator.fermion.number`

Each operator takes a site and spin projection ( `sz` ) in order to find the right position in the Hilbert space samples. We will create a helper function to abbreviate the creation, annihilation and number operators in the example below.

```python
from netket import experimental as nkx
from netket.experimental.operator.fermion import (
    create as c, destroy as cdag, number as nc)

# create the graph our fermions can hop on
L = 4
g = nk.graph.Hypercube(length=L, n_dim=2, pbc=True)
n_sites = g.n_nodes

# create a hilbert space with 2 up and 2 down spins
hi = nkx.hilbert.SpinOrbitalFermions(n_sites, s=1 / 2, n_fermions=(2, 2))

t = 1     # tunneling/hopping
U = 0.01  # coulomb

up, down = +0.5, -0.5
ham = 0.0
for sz in (up, down):
    for u, v in g.edges():
        ham += -t * (cdag(hi, u, sz) * c(hi, v, sz) +
                     cdag(hi, v, sz) * c(hi, u, sz))
for u in g.nodes():
    ham += U * nc(hi, u, up) * nc(hi, u, down)
```

**Sampling:** To run a VMC optimization, we need a proper sampling algorithm that takes into account the constraints of the computational basis we are working with. As the `SpinOrbitalFermions` basis consereves total spin-magnetization, we cannot use samplers like `MetropolisLocal` which randomly change the population $n_{i,\sigma}$ on a site, thus changing the total spin. We can instead use `MetropolisExchange`, which moves fermions around according to the physical lattice graph of $L \times L$ vertices, but the computational basis defined by the Hilbert space contains $(2s+1)L^2$ occupation numbers. By taking a disjoint copy of the lattice, we can move the fermions around independently for both spins and therefore conserve the number of fermions with up and down spin. Notice that in the chosen representation, where is no need to anti-symmetrize our ansatz.

```python
sa = nk.sampler.MetropolisExchange(hi,
    graph=nk.graph.disjoint_union(g, g),
    n_chains=16)

ma = nk.models.RBM(alpha=1, param_dtype=complex)
vs = nk.vqs.MCState(sa, ma, n_discard_per_chain=100, n_samples=512)

opt = nk.optimizer.Sgd(learning_rate=0.01)
sr = nk.optimizer.SR(diag_shift=0.1)

gs = nk.driver.VMC(ham, opt, variational_state=vs, preconditioner=sr)
gs.run(500, out='fermions')
```

# 9    Example: Real-time dynamics

We demonstrate the simulation of NQS dynamics in the transverse-field Ising model (TFIM) on an $L$ site chain with periodic boundaries, using a restricted Boltzmann machine (RBM) as the NQS ansatz. The Hamiltonian reads

$$\hat{H}_{\text{Ising}} = J \sum_{\langle ij \rangle} \hat{\sigma}_i^z \hat{\sigma}_j^z - h \sum_i \hat{\sigma}_i^x \tag{54}$$

with $J = 1$ and $h = 1$ and periodic boundary conditions. We will estimate the expectation value of the transverse magnetization $\hat{S}^x = \sum_i \hat{\sigma}_i^x$ along the way.

We simulate the dynamics starting from an initial state $|\psi(t_0)\rangle$ that is the ground state for the TFIM Hamiltonian with $h = 1/2$. The random weight initialization of a neural network yields a random initial state. Therefore, we determine the weights corresponding to this initial state by performing a ground-state optimization. Even though the TFIM ground state can be parametrized using an RBM ansatz with real-valued weights, we need to use complex-valued weights here, in order to describe the complex-phase of the wave function that arises during the time evolution[21]. In this example, we work with a chain of $L = 20$ sites, which can be easily simulated on a typical laptop with the parameters below.

```python
import netket as nk
import netket.experimental as nkx

# Spin Hilbert space on 20-site chain with PBC
L = 20
chain = nk.graph.Chain(L)
hi = nk.hilbert.Spin(s=1/2) ** L

# Define RBM ansatz and variational state
rbm = nk.models.RBM(alpha=1, param_dtype=complex)
sampler = nk.sampler.MetropolisLocal(hi)
vstate = nk.vqs.MCState(sampler, rbm, n_samples=4096)

# Hamiltonian at J=1 (default) and external field h=1/2
ha0 = nk.operator.Ising(hi, chain, h=0.5)
# Observable (transverse magnetization)
obs = {"sx": sum(nk.operator.spin.sigmax(hi, i) for i in range(L))}

# First, find the ground state of ha0 to use it as initial state
optimizer = nk.optimizer.Sgd(0.01)
sr = nk.optimizer.SR()
vmc = nk.VMC(ha0, optimizer, variational_state=vstate, preconditioner=sr)
# We run VMC with SR for 300 steps
vmc.run(300, out="ising1d_groundstate_log", obs=obs)
```

## 9.1    Unitary dynamics

Starting from the ground state, we can compute the dynamics after a quench of the transverse field strength to $h = 1$. We use the second-order Heun scheme for time stepping, with a step size of $dt = 0.01$, and explicitly specify a QGT implementation (compare Sec. 3.5) in order to make use of the more efficient code path for holomorphic models.

---

[21]It is possible to first use a real-weight RBM for the initial state preparation and then switch to complex-valued weights for the dynamics. For the sake of simplicity, we leave out this extra step in the present example.

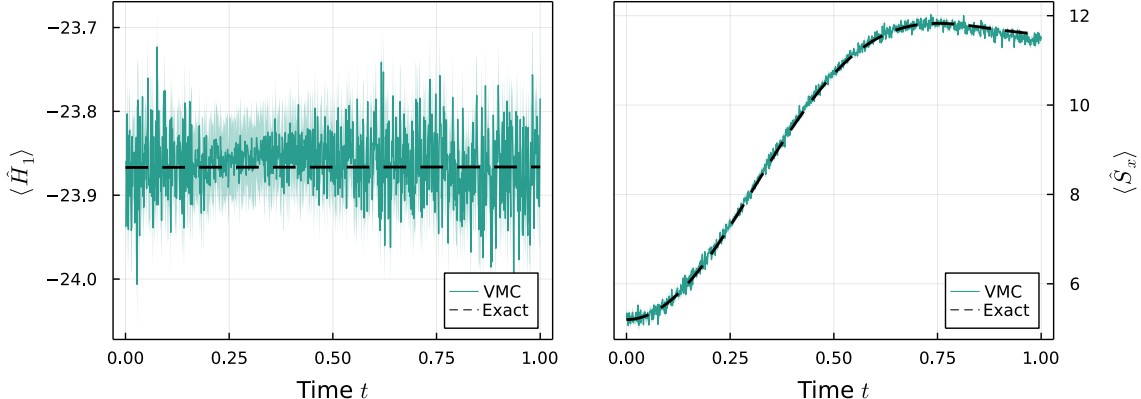

Figure 4: Comparison between the exact (dashed line) and variational dynamics of a quench on the transverse-field Ising model. **(Left):** Expectation value of the quenched Hamiltonian, which is conserved by the unitary dynamics. The shaded area represents the uncertainty due to Monte Carlo sampling. **(Right):** Expectation value of the total magnetization along the $x$ axis during the evolution.

```
1   # Quenched Hamiltonian
2   ha1 = nk.operator.Ising(hi, chain, h=1.0)
3   # Heun integrator configuration
4   integrator = nkx.dynamics.Heun(dt=0.001)
5   # QGT options
6   qgt = nk.optimizer.qgt.QGTJacobianDense(holomorphic=True)
7   # Creating the time-evolution driver
8   te = nkx.TDVP(ha1, vstate, integrator, qgt=qgt)
9   # Run the t-VMC solver until time T=1.0
10  te.run(T=1.0, out="ising1d_quench_log", obs=obs)
```

In Fig. 5 we show the results of this calculation, comparing against an exact solution computed using QuTiP [90, 91].

## 9.2 Dissipative dynamics (Lindblad master equation)

In Section 4.3 we have shown that the time-dependent variational principle can also be used to study the dissipative dynamics of an open quantum system. In this section we give a concrete example, studying the transverse-field Ising model coupled to a zero-temperature bath. The coupling is modeled through the spin depolarization operators $\hat{\sigma}_i^-$ acting on every site $i$.

As the dissipative dynamics converges to the steady state, which is also an attractor of the non-unitary dynamics, we will use a weak-simulation of the dynamics[22] to determine the steady-state more efficiently than using the natural gradient descent scheme proposed in ref. [25]. This scheme is similar to what was proposed in Ref. [26].

We employ the positive-definite RBM ansatz proposed in Ref. [52]. A version of that network with complex-valued parameters is provided in NETKET with the name `nk.models.NDM`.

```
1   # The graph of the Hamiltonian
```

---

[22]We use weak and strong simulation in the sense of the theory of numerical SDE schemes [92]. This means that weak integration is an integration which is not accurate at finite times but which converges to the right state at long times. A strong integration yields the correct state at every time $t$.

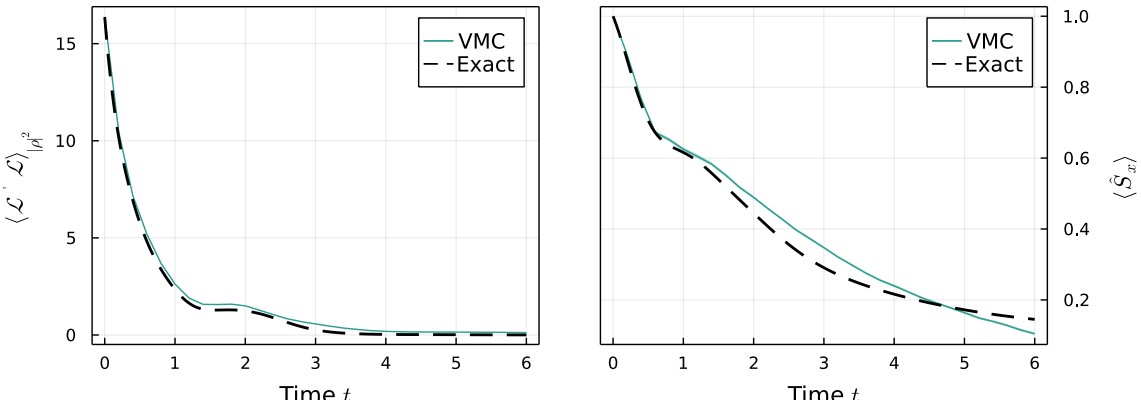

Figure 5: Comparison between the exact (dashed line) and variational dynamics of a random initial density matrix evolved according to the Lindblad Master equation. **(Left):** Expectation value of the $\langle\langle\rho|\mathcal{L}^\dagger\mathcal{L}|\rho\rangle\rangle$ convergence estimator. **(Right):** Expectation value of the total magnetization along the $\hat{x}$ axis during the evolution. We remark that the evolution is near-exact in the region where the dissipative terms dominate the dynamics, while there is a sizable error when the unitary dynamics starts to play a role. The error could be reduced by considering smaller time steps.

```
2  g = nk.graph.Chain(L, pbc=False)
3  # Hilbert space
4  hi = nk.hilbert.Spin(0.5)**g.n_nodes
5  # The Hamiltonian
6  ha = nk.operator.Ising(hi, graph=g, h=-1.3, J=0.5)
7  # Define the list of jump operators
8  j_ops = [nk.operator.spin.sigmam(hi, i) for i in range(g.n_nodes)]
9  # Construct the Liouvillian
10 lind = nk.operator.LocalLiouvillian(ha, j_ops)
11
12 # observable
13 Sx = sum([nk.operator.spin.sigmax(hi, i) for i in
       range(g.n_nodes)])/g.n_nodes
14
15 # Positive-definite RBM-like ansatz (Torlai et al.)
16 ma = nk.models.NDM(alpha=2, beta=3)
17 # MetropolisLocal sampling on the Choi's doubled space.
18 sa = nk.sampler.MetropolisLocal(lind.hilbert, n_chains=16)
19 # Mixed Variational State. Use less samples for the observables.
20 vs = nk.vqs.MCMixedState(
21     sa, ma, n_samples=12000, n_samples_diag=1000, n_discard_per_chain=100
22 )
23 # Setup the ODE integrator and QGT.
24 integrator = nkx.dynamics.Heun(dt=0.01)
25 # The NDM ansatz is not holomorphic because it uses conjugation
26 qgt = nk.optimizer.qgt.QGTJacobianPyTree(holomorphic=False, diag_shift=1e-3)
27 te = nkx.TDVP(lind, variational_state=vs, integrator=integrator, qgt=qgt)
28
29 # run the simulation and compute observables
30 te.run(T=6.0, obs={"Sx": Sx, "LdagL": lind.H @ lind})
```

In the listing above, we first construct the Liouvillian by assembling the Hamiltonian

and the jump operators, then we construct the variational mixed state. We chose a different number of samples for the diagonal, used when sampling the observables, as that happens on a smaller space with respect to the full system. For the geometric tensor, we choose the `QGTJacobianPyTree` and specify that the ansatz is non-holomorphic (while this is already the default, a warning would be printed otherwise, asking the user to be explicit). The choice is motivated by the fact that the TDVP driver by default uses an SVD-based solver, which works best `QGTJacobian`-based implementations. However, `NDM` uses a mix of complex and real parameters which is not supported by `QGTJacobianDense`, and would throw an error. Normally, to simulate a meaningful dynamics you'd want to keep the diagonal shift small, but since we are striving for a weak simulation a large value helps stabilize the dynamics.

# 10 Benchmarks

## 10.1 Variational Monte Carlo

We benchmark NETKET by measuring its performance on the 1D/2D transverse-field Ising model defined as in Eq. (54) with $J = 1$ and $h = 1$ and periodic boundary conditions.

    We first monitor the scaling behavior of NETKET's VMC implementation by running an energy optimization consisting of 100 steps. In order to carry out a meaningful benchmark, we run a first VMC step to trigger the JIT-compiler on the relevant JAX and NUMBA functions, while all reported timings are for the evaluation of JIT compiled functions only. The left panel of Fig. 6 depicts the scaling behavior of the computational time as a function of the complexity of the NQS model, using three implementations of the QGT. Hereby, we increase the complexity of the NQS by optimizing a DNN with an increasing amount of layers, where depth $d$ represents the number of dense layers (with $\alpha = 1$), each followed by a sigmoid activation function. Such a DNN with $d$ layers has $\mathcal{O}((d-1)L^2 + L)$ free parameters.

## 10.2 MPI for NetKet

We benchmark the scaling behavior of NETKET as a function of the computational resources available to perform parallel computations. Therefore, NETKET uses MPI for JAX through MPI4JAX [38]. The effectiveness of the MPI implementation is illustrated in Fig. 7 for a VMC optimization with and without Stochastic Reconfiguration (SR). Throughout our analyses, we provide each MPI rank with 2 CPU cores. We introduce the speedup factor $\tau_r = \Delta t_1 / \Delta t_r$ where $\Delta t_r$ refers to the time required to perform the computations on $r$ MPI ranks. Similarly, we define $\tau_n$ as the speedup factor when using $n$ nodes. The right panel of Fig. 7 demonstrates that even on a single node, our MPI implementation can introduce significant speedups by running multiple Markov Chain samplers in parallel. This is consistent with the fact that JAX is not able to make use of multiple CPU cores unless working on very large matrices.

    The performance of NETKET on challenging Hamiltonians, as well as its scalability with both system size and model complexity depends on the implementation details of the quantum geometric tensor [see Eq. (22)] and its matrix multiplication with the gradient vector, as discussed in Section 3.5. We therefore isolate these operations and benchmark the QGT constructor, combined with 1000 matrix multiplications with the gradient vector (where the latter imitates many steps in the iterative solver). In Fig. 7, we show the scaling behavior of these operations with respect to both the number of ranks (on a

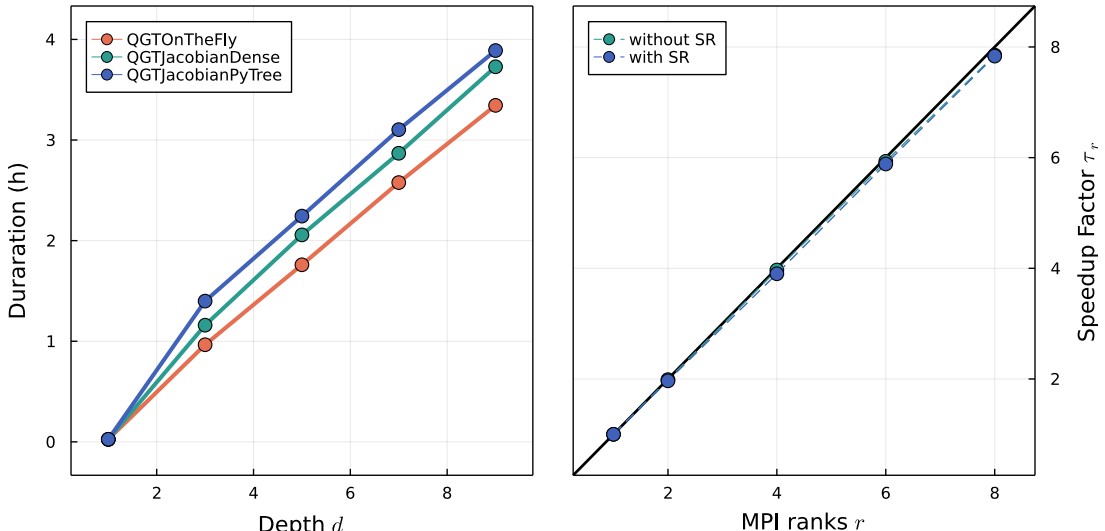

Figure 6: **(Left):** Benchmark of NETKET's VMC implementation. Each data point shows the minimum time spent (out of 5 repetitions) to evolve a DNN with depth $d$ layers and complex weights over 100 VMC steps for the 1D transverse-field Ising model with $L = 256$ and $N_{\text{samples}} = 2^{14} = 16384$. **(Right):** Scaling behavior of the required computational time as a function of the number of MPI ranks on a single node. We repeat 5 VMC optimizations and report the minimum time required for 100 steps for the 1D Ising model with $L = 256$ and an RBM with $\alpha = 1$ with complex weights. The black line represents ideal scaling behavior.

single node), and its scaling behavior with respect the number of nodes (thereby including communication over the Infiniband network). One can observe that the scaling behavior is close to optimal, especially when the number of samples per rank is sufficiently large.

## 10.3 Comparison with jVMC

We compare NETKET with jVMC [40], another open-source Python package supporting VMC optimization of NQS. Results are shown in Fig. 6. We remark that since both NETKET and jVMC are JAX-based, performance on sampling, expectation values and gradients is roughly equivalent when using the same hyperparameters [40]. However, a performance difference arises in algorithms requiring the use of the QGT, such as TDVP or natural gradient (SR). Such difference will vanish in cases where the cost of solving the QGT linear system is small with respect to the cost of computing the energy and its gradient.

At the time of writing, jVMC only implements a singular-value decomposition (SVD) solver to invert the QGT matrix. The same type of solver can be used also in NETKET (for a detailed discussion, see Section 3.5). We limit the computations to models with less than 7000 parameters in order for the QGT to have less than $49 \cdot 10^3$ elements, which is approximately the maximum matrix size that can be diagonalized in reasonable time on the GPUs we have access to[23]. For that reason we chose the size of $L = 64$ spins.

As shown in Table 5, NETKET outperforms by almost an order of magnitude jVMC on a 32-core CPU using SVD-based solvers. On GPU, jVMC requires significantly less

---

[23]Using distributed linear-algebra libraries such as ScaLaPack [93], ELPA [94] or the recent [95] would allow us to avoid this barrier, however we are not aware of any Python binding for those libraries. Regardless, if those libraries exposed a distributed linear solver to Python, using it with NETKET would be as simple as using it as the linear solver for the QGT.

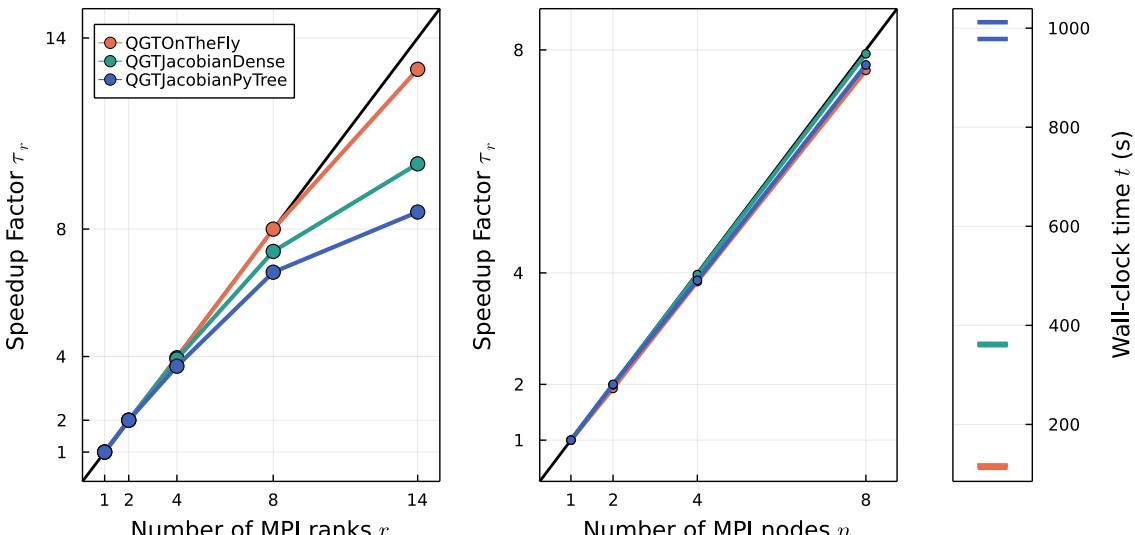

Figure 7: **(Left):** Speedup factor $\tau$ observed by increasing the number of MPI ranks $r$ on a single node while keeping the problem size constant (strong scaling). **(Center):** Speedup factor observed by scaling across multiple nodes $n$, and scaling the number of samples accordingly (weak scaling). **(Right):** Scatter plot of the absolute wall clock time in seconds for the runs reported in the weak scaling (center) plot. There are 4 points for every color, but they overlap for most implementations because of the almost-ideal weak scaling. We repeat 5 iterations of constructing the QGT and 1000 matrix multiplications with the gradient vector for the 2D Ising model with $L = 8$ and a DNN with 9 layers and complex weights. In the left panel, we keep the number of samples $N_{samples} = 2^{14}$ constant, while in the center panel, we increase the number of samples to $2^{14} \times n$, while we correct the timing by the number of nodes $n$ to show the speedup factor for a constant number of $2^{14}$ samples. We remark that while the speedup factor is resistant to changes in the network architecture, the absolute timing might favor one or another implementation depending on several details and can change depending on the architecture and problem at hand.

computational time than on CPU, yet, NETKET outperforms jVMC by about 50% in a full VMC iteration. We remark that in this benchmark both packages scale poorly when going from a single GPU to two. This is because the diagonalization of the QGT, in this case the bottleneck, cannot be parallelized.

In order to scale efficiently to many GPUs, our QGT implementations can be combined with iterative solvers to scale up to potentially millions of parameters, as well as significantly larger system sizes. As expected from Fig. 7, increasing the number of MPI ranks reduces the total time by a factor nearing the number of ranks (with eventually a saturation in the speedup). Notice also that the CG-solver becomes significantly more efficient on GPU.

## 11 Discussion and conclusion

We have presented NETKET 3, a modular Python toolbox to study complex quantum-mechanical problems with machine learning-inspired tools. Compared to version 2 [36], the

|  |  | **NetKet** | **jVMC** |
|---|---|---|---|
| **SVD solver** | CPU (32 cores) | 48 | 337 |
|  | GPU (×1) | 24 | 44 |
|  | GPU (×2) | 20 | 36 |
| **CG solver** | CPU (32 cores) | 86 | N/A |
|  | CPU (32 cores, MPIx16) | 7.5 | N/A |
|  | GPU (×1) | 3.9 | N/A |
|  | GPU (×2) | 1.7 | N/A |

Table 5: Comparison of performance between NETKET and jVMC. All times are indicated in seconds and have been taken on a workstation with an `AMD Ryzen Threadripper 3970X 32-Core` processor and `2xNvidia RTX 3090` GPUs. Timings are for one VMC step using a complex-valued RBM model with hidden unit density of $\alpha = N_{\text{hidden}}/L = 1$ on the 1-dimensional TFIM model (54) with $L = 64$ sites. Other parameters are: $2^{14}$ total samples, $2^{10}$ independent Markov chains per GPU (or across all CPUs). Calculations for NETKET where performed using `QGTJacobianDense(holomorphic=True)`. NETKET multi-GPU calculations use CUDA-enabled MPI for inter-process communication, while jVMC uses JAX built-in mechanism. The run labeled with (MPI×16) is run on the same workstation but 16 MPI processes are used to better take advantage of the multiple cores of the processors.

major new feature is the ability to define arbitrary neural-network ansätze for either wave functions or density matrices using the flexible JAX framework; we believe this makes NETKET much more useful to non-technical users. Another significant improvement is that NETKET is now completely modular. Users of NETKET 3 can decide to use only the neural-network architectures, the stochastic samplers, the quantum geometric tensor, or the operators without necessarily requiring the `VariationalState` interface, which is convenient but geared towards the most common applications of variational NQS. Care has been taken to ensure that the algorithms implemented can scale to very large systems and models with millions of parameters thanks to more efficient implementations of the geometric tensor and other algorithmic bottlenecks. Thanks to its JAX foundations, NETKET 3 can now also make effective use of GPU hardware, without any need for manual low-level programming for these platforms.

Even with all the new features that have been introduced with this version, there are many things that we would like to see integrated into NETKET in the future. To name a few: native support for fermionic systems; support for more general geometries in continuous systems; improvements to the dynamics submodule in order to support a wider variety of ODE solvers and more advanced regularization and diagnostic schemes [22, 24]; new drivers for quantum state reconstruction [32] and overlap optimization [33]; a more general way to define arbitrary cost functions to be optimized. However, we think that our new JAX-based core is very welcoming to contributors, and we believe that this constitutes a solid foundation upon which to build in the future. Moreover, we are now explicitly committed to stability of the user-facing API, in order to make sure that code written today will keep working for a reasonable time, while we iterate and refine NETKET.

Since a project is only as big as its community, the most important developments are probably those related to documentation, learning material, and developing a community

where users answer each other's questions in the spirit of open, shared science. We are taking steps to make all of this happen.

# Acknowledgments

We are grateful to the authors and maintainers of JAX [5], FLAX [45], and OPTAX [66] for guidance and constructive discussions on how to best use their tools. F.V. thanks W. Bruinsma for adapting plum-dispatch to our needs and making life more Julian in the Python world. We also thank all the users who engaged with us on GitHub reporting bugs, sharing ideas, and ultimately helping us to produce a better framework.

**Funding information.** This work is supported by the Swiss National Science Foundation under Grant No. 200021_200336. A. Sz. gratefully acknowledges the ISIS Neutron and Muon Source and the Oxford–ShanghaiTech collaboration for support of the Keeley–Rutherford fellowship at Wadham College, Oxford.

# A  Details of group convolutions

## A.1  Group convolutions and equivariance

As explained in Section 5.2, GCNNs generate the action of every element $g$ of the symmetry group $G$ on a wave function $|\psi\rangle$, written as $|\psi_g\rangle = g|\psi\rangle$, which are then combined into a symmetric wave function using the projection operator (40). Amplitudes of the computational basis states $|\boldsymbol{\sigma}\rangle$ are related to one another in these wave functions as

$$\psi_g(\boldsymbol{\sigma}) = \langle\boldsymbol{\sigma}|g|\psi\rangle = \psi(g^{-1}\boldsymbol{\sigma}). \tag{55}$$

Therefore, feature maps inside the GCNN are indexed by group elements rather than lattice sites, and all layers must be *equivariant:* that is, if their input is transformed by a space-group symmetry, their output must be transformed the same way, so that (55) would always hold. Pointwise nonlinearities clearly fit this bill [74]; we now consider what linear transformations are allowed.

First, input feature maps naturally defined on lattice sites must be embedded into group-valued features:

$$\mathbf{f}_g = \sum_{\vec{r}} \mathbf{W}_{g^{-1}\vec{r}}\sigma_{\vec{r}} = \sum_{\vec{r}} \mathbf{W}_r \sigma_{g\vec{r}} = \sum_{\vec{r}} \mathbf{W}_{\vec{r}}(g^{-1}\boldsymbol{\sigma})_{\vec{r}}, \tag{56}$$

consistent with (55). We also see that the embedding (56) is equivariant: if the input is transformed by some symmetry operation $u$, the output transforms as

$$\sum_{\vec{r}} \mathbf{W}_{g^{-1}\vec{r}}\sigma_{u\vec{r}} = \sum_{\vec{r}} \mathbf{W}_{g^{-1}u^{-1}\vec{r}}\sigma_{\vec{r}} = \mathbf{f}_{ug}. \tag{57}$$

This layer is implemented in NETKET as `nk.nn.DenseSymm`.

To build deeper GCNNs, we also need to map group-valued features onto one another in an equivariant fashion. This is achieved by *group convolutional* layers, which transform input features as[24]

$$\phi_g = \sum_{h \in G} \mathbf{W}_{h^{-1}g}\mathbf{f}_h. \tag{58}$$

---

[24]Our convention differs from that of Ref. [74], which in fact implements group correlation rather than convolution. The two conventions are equivalent (the indexing of the kernels differs by taking the inverse of each element); we use convolutions to simplify the Fourier transform-based implementations of Sec. A.2.

This layer is implemented in NETKET as `nk.nn.DenseEquivariant`. It is equivariant under multiplying with a group element $u$ from the left:

$$\sum_{h \in G} \mathbf{W}_{h^{-1}g} \mathbf{f}_{uh} = \sum_{h \in G} \mathbf{W}_{h^{-1}ug} \mathbf{f}_h = \phi_{ug}, \tag{59}$$

which is consistent with how the embedding layer (56) is equivariant, cf. (57). Indeed, it can be composed with (56):

$$\phi_g = \sum_h \mathbf{W}_{h^{-1}g} \sum_{\vec{r}} \mathbf{W}'_{h^{-1}\vec{r}} \sigma_{\vec{r}} = \sum_{\vec{r}} \left( \sum_h \mathbf{W}_{h^{-1}} \mathbf{W}'_{h^{-1}g^{-1}\vec{r}} \right) \sigma_{\vec{r}} \equiv \sum_{\vec{r}} \mathbf{W}''_{g^{-1}\vec{r}} \sigma_{\vec{r}} \tag{60}$$

as the expression in brackets only depends on $g^{-1}\vec{r}$.

Finally, the output features of the last layer of the GCNN are turned into the wave functions $\psi_g(\boldsymbol{\sigma}) = \sum_i \exp\left(f_g^{(i)}\right)$ and projected on an irrep using (40) (we drop the irrelevant constant prefactor):

$$\psi(\boldsymbol{\sigma}) = \sum_{i,g} \chi_g^* \exp\left(f_g^{(i)}\right), \tag{61}$$

where $\chi_g$ are the characters of the irrep. In addition to allowing nontrivial symmetries, our choice of summing a large number of terms in the ansatz appears to improve the stability of variational optimization for sign-problematic Hamiltonians [16, 31, 50, 73].

## A.2 Fast group convolutions using Fourier transforms

The simplest implementation of a group convolutional layer is expanding each of the $f_{\rm in} f_{\rm out}$ kernels, containing $|G|$ entries, to a $|G| \times |G|$ matrix defined as

$$\tilde{W}_{g,h}^{(a,b)} = W_{g^{-1}h}^{(a,b)}, \tag{62}$$

where $a, b$ index input and output features, respectively. The resulting tensor of size $f_{\rm in} f_{\rm out} |G|^2$ can then be contracted straightforwardly with the input features:

$$\phi_h^{(b)} = \sum_{a,g} \tilde{W}_{g,h}^{(a,b)} f_g^{(a)} \tag{63}$$

is equivalent to (58). Embedding layers (56) can be constructed analogously. This method is the easiest to interpret and code, serving as a useful check on other methods; however, the enlarged kernels require substantial amounts of memory, which already becomes a serious problem on modestly sized lattices and networks. Furthermore, evaluating a convolution using this method takes $O(f_{\rm in} f_{\rm out} |G|^2)$ time. NETKET implements two approaches to improve on this scaling.

The first approach uses *group Fourier transforms,* which generalize the usual discrete Fourier transform for arbitrary finite groups. The forward and backward transformations are defined by

$$\hat{f}(\rho) = \sum_{g \in G} f(g)\rho(g); \qquad f(g) = \frac{1}{|G|} \sum_\rho d_\rho \operatorname{Tr}\left[\hat{f}(\rho)\rho(g^{-1})\right]. \tag{64}$$

In the forward transformation, $\rho$ is a representation of the group $G$; $f(g)$ is a function defined on group elements, while $\hat{f}(\rho)$ is a matrix of the same shape as the representatives $\rho(g)$. The sum in the backward transformation runs over all inequivalent irreps $\rho$, of dimension $d_\rho$, of the group. Since $\sum_\rho d_\rho^2 = |G|$, this transformation does not increase the

amount of memory needed to store inputs, outputs, or kernels.[25] Group convolutions can readily be implemented by multiplying the Fourier transform matrices (we drop feature indices for brevity):

$$\hat{\phi}(\rho) = \sum_g \phi_g \rho(g) = \sum_{g,h} f_h W_{h^{-1}g} \rho(h) \rho(h^{-1}g) = \hat{f}(\rho) \hat{W}(\rho). \tag{65}$$

To calculate a convolution using this approach, the input features are Fourier transformed $[O(f_{\text{in}}|G|^2)$ as there is no generic fast Fourier transform algorithm for group Fourier transforms], multiplied with the kernel Fourier transform for each irrep $[O(f_{\text{in}}f_{\text{out}}d_\rho^3)$ for an irrep of dimension $d_\rho]$, and the output is transformed back $[O(f_{\text{out}}|G|^2)]$, yielding the total runtime $O[(f_{\text{in}} + f_{\text{out}})|G|^2 + f_{\text{in}}f_{\text{out}}\sum_\rho d_\rho^3]$. In a large space group, most irreps are defined on a star of $|P|$ wave vectors ($P$ is the point group) and thus have dimension $|P|$; accordingly, $\sum_\rho d_\rho^3 \approx |G||P|$.

The second approach, based on Ref. [74], exploits the fact that the translation group $T$ is a normal subgroup of the space group $G$, so each $g \in G$ can be written as $t_g p_g$, where $t_g$ is a translation and $p_g$ is a fixed coset representative (in symmorphic groups, we can choose these to be point-group symmetries). Now, we can define the expanded kernels (we drop feature indices again to reduce clutter)

$$\tilde{W}_t^{(p_g,p_h)} \equiv W_{p_g^{-1}tp_h} \tag{66}$$

such that

$$\phi_h = \sum_g f_g W_{g^{-1}h} = \sum_{t_g,p_g} f_{t_g p_g} W_{p_g t_g^{-1} t_h p_h} \equiv \sum_{t_g,p_g} \tilde{f}_{t_g}^{(p_g)} \tilde{W}_{t_h-t_g}^{(p_g,p_h)}. \tag{67}$$

In the last form, we split the space-group feature map $f$ into cosets of the translation group and observe that the latter is Abelian. In fact, the translation group is equivalent to the set of valid lattice vectors, so the sum over $t_g$ in (67) is a standard convolution. Ref. [74] proposes to perform this convolution using standard cuDNN routines. However, we are usually interested in convolutions that span the entire lattice in periodic boundary conditions: these can be performed more efficiently using fast Fourier transforms (FFTs) as the Fourier transform of a convolution is the product of Fourier transforms. Therefore, we FFT the kernels $\tilde{W}$ and features $\tilde{f}$, contract them as appropriate, and FFT the result back:

$$\tilde{\phi}^{(b,p_h)} = \mathcal{F}^{-1}\left[ \sum_{a,p_g} \mathcal{F}\left(\tilde{f}^{(a,p_g)}\right) \mathcal{F}\left(\tilde{W}^{(a,b;p_g,p_h)}\right) \right], \tag{68}$$

where the Fourier transform is understood to act on the omitted translation-group indices, and the Fourier transforms are multiplied pointwise.

Calculating a convolution in this approach involves $f_{\text{in}}|P|$ forward FFTs $[O(|T|\log|T|)$ each], $|T|$ tensor inner products $[O(f_{\text{in}}f_{\text{out}}|P|^2$ each], and $f_{\text{out}}|P|$ backward FFTs; as $|G| = |T||P|$, this yields a total of $O[(f_{\text{in}} + f_{\text{out}})|G|\log|T| + f_{\text{in}}f_{\text{out}}|G||P|]$. For large lattices, which bring out the better asymptotic scaling of FFTs, this improves significantly on the runtime of the group Fourier transform-based approach, especially in the pre- and postprocessing stages. By contrast, the group Fourier transform approach is better for large point groups, as it avoids constructing the $|P|^2$ reshaped kernels $\tilde{W}^{p_g p_h}$, which can be prohibitive for large lattices.

In practice, as both FFTs and group Fourier transforms involve steps more complicated than simple tensor multiplication, their performance is hard to assess beyond asymptotes,

---

[25]If some irreps cannot be expressed as matrices with real entries, the Fourier transform of real inputs/outputs/kernels is complex too, temporarily doubling the amount of memory used.

especially on a GPU. On CPUs, the FFT-based approach tends to be faster. On GPUs, computation time tends to scale sub-linearly with the number of operations so long as the process is efficiently parallelized. As all operations of the group Fourier transform implementation involve multiplications of large matrices, it can fully exploit the large GPU registers even with relatively few samples. By contrast, FFTs cannot be fully vectorized, meaning that larger batches are required to make full use of the computing power of the GPU. In practice therefore, the FFT-based approach may not perform better until most of the GPU memory becomes involved in evaluating a batch.

# B    Implementation details of the quantum geometric tensor

In the following, we discuss our implementations of the quantum geometric tensor, introduced in section 3.5, in more detail. In particular, we show how the action of the quantum geometric tensor on a vector can be computed efficiently without storing the full matrix. Appendix B.1 introduces relevant automatic-differentiation concepts in general terms; the concrete algorithms used by `QGTJacobian` and `QGTOnTheFly` are discussed in Appendices B.2 and B.3, respectively.

## B.1    Jacobians and their products

We assume that our NQS is modeled by the scalar parametric function $f(s) = \ln \psi_\theta(s)$, where $\theta$ is a vector of variational parameters and $s$ is a basis vector of the Hilbert space. Consistent with the notation of the main text, $O_j(s) = \partial_{\theta_j} \ln \psi_\theta(s)$ are the log-derivatives (13) of the NQS.

We also assume that $f$ can be vectorized and evaluated for a batch of inputs $\{s_k\}_{k=1\ldots N_s}$, yielding the vector $f_k = \ln \psi_\theta(s_k)$. The Jacobian of this function is therefore the matrix

$$J_{kl} = O_l(s_k) = \frac{\partial \ln \psi_\theta(s_k)}{\partial \theta_l}; \tag{69}$$

each row corresponds to the gradient of $f$ evaluated at a different input $s_k$, so $k = 1 \ldots N_s$ and $l = 1 \ldots N_{\text{parameters}}$.

The Jacobian matrix can be computed in JAX with `jax.jacrev(log_wavefunction)(s)`, which returns a matrix.[26] However, it is often not needed to have access to the full Jacobian: for example, when computing the gradient (26) of the variational energy, we only need the product of the Jacobian with a vector, namely $\Delta E_{\text{loc}}(s_k) = \tilde{H}(s_k) - \mathbb{E}[\tilde{H}]$.

A vector can be contracted with the Jacobian along its dimension corresponding to either parameters or outputs:

- *Jacobian–vector products* (Jvp), $\tilde{\mathbf{v}} = J\mathbf{v}$, can be computed using forward-mode automatic differentiation;

- *vector–Jacobian products* (vJp), $\tilde{\mathbf{v}} = \mathbf{v}^T J$, can be computed through backward-mode automatic differentiation (backward propagation).

Modern automatic-differentiation frameworks like that of JAX implement primitives that evaluate Jvp and vJp, and construct higher-level functions such as `jax.grad` or `jax.jacrev` on top of those functions; that is, one can extract the best performance from JAX by making use of vJp and Jvp as much as possible [96].

---

[26]More precisely, it returns a PyTree with a structure similar to the PyTree that stores the parameters; each leaf gains an additional dimension of length $N_s$.

## B.2  QGTJacobian

Writing the estimator (23) of the quantum geometric tensor explicitly in terms a finite number of samples $s_k$, we obtain

$$
\begin{aligned}
G_{ij} &= \mathbb{E}\left[O_i^* O_j\right] - \mathbb{E}\left[O_i\right]^* \mathbb{E}\left[O_j\right] \\
&\approx \frac{1}{N_s} \sum_{k=1}^{N_s} O_i(s_k)^* O_j(s_k) - \frac{1}{N_s^2} \left( \sum_{k=1}^{N_s} O_i(s_k) \right)^* \left( \sum_{k=1}^{N_s} O_j(s_k) \right) \\
&= \frac{1}{N_s} \sum_{k=1}^{N_s} \left( O_i(s_k) - \sum_{k=1}^{N_s} \frac{O_i(s_k)}{N_s} \right)^* \left( O_j(s_k) - \sum_{k=1}^{N_s} \frac{O_j(s_k)}{N_s} \right) \\
&= \frac{1}{N_s} \sum_{k=1}^{N_s} \left( J_{ki} - \sum_{k=1}^{N_s} \frac{J_{ki}}{N_s} \right)^* \left( J_{kj} - \sum_{k=1}^{N_s} \frac{J_{kj}}{N_s} \right) \\
&= \frac{1}{N_s} \sum_{k=1}^{N_s} \left( \Delta J_{ki} \right)^* \left( \Delta J_{kj} \right),
\end{aligned}
\tag{70}
$$

where we have defined the centered Jacobian $\Delta J_{ki} \equiv J_{ki} - \sum_{k=1}^{N_s} \frac{J_{ki}}{N_s}$. In matrix notation, this is equivalent to

$$
G = \frac{\Delta J^\dagger}{\sqrt{N_s}} \frac{\Delta J}{\sqrt{N_s}}.
\tag{71}
$$

The Jacobian-based implementation of the quantum geometric tensor computes[27] and stores[28] the full Jacobian matrix $J_{kl}$ for the given samples upon construction. Then, QGT–vector products $\tilde{\mathbf{v}} = G\mathbf{v}$ are computed without finding the full matrix $G$, in two steps:

$$
\Delta \mathbf{w} = \frac{\Delta J}{\sqrt{N_s}} \mathbf{v}; \qquad\qquad \tilde{\mathbf{v}} = \frac{\Delta J^\dagger}{\sqrt{N_s}} \Delta \mathbf{w} = \left( \frac{\Delta J}{\sqrt{N_s}} \Delta \mathbf{w}^\dagger \right)^\dagger;
\tag{72}
$$

the final form has the advantage that the Hermitian transpose of a vector is simply its conjugate. Evaluating Eq. (72) is usually less computationally expensive than constructing the full quantum geometric tensor.

## B.3  QGTOnTheFly

In some cases one might have so many parameters or samples that it is impossible to store the full Jacobian matrix in memory. In that case, we still evaluate a set of equations similar to Eq. (72), but without pre-computing the full Jacobian, only using vector–Jacobian and Jacobian–vector products.

It would be impractical to perform a vJp using the centered Jacobian; however, Eq. (23) can be rewritten as $G_{ij} = \mathbb{E}\left[O_i^* \left(O_j - \mathbb{E}\left[O_j\right]\right)\right]$, which yields

$$
G = \frac{1}{N_s} J^\dagger \Delta J,
\tag{73}
$$

where we have substituted one of the two centered Jacobians with a plain Jacobian. Then, we note that the centered-Jacobian–vector product can be expressed as

$$
\Delta \mathbf{w} \equiv \Delta J \, \mathbf{v} = \left( J - \frac{\mathbf{1}^T J}{N_s} \right) \mathbf{v} = \mathbf{w} - \frac{\mathbf{1}^T \mathbf{w}}{N_s},
\tag{74}
$$

---

[27]The full Jacobian can be computed row by row, performing vJps with vectors that have a single nonzero entry. In practice, as the network is evaluated independently for all samples in a batch, each of these products requires us to back-propagate the network for only one sample at a time.

[28]To speed up the evaluation of (72), we actually store $\Delta J / \sqrt{N_s}$.

where $\mathbf{1}^T$ is a row vector all entries of which are 1, used to express averaging the columns of the Jacobian in the matrix formalism. Therefore, `QGTOnTheFly` performs the following calculations:

$$\mathbf{w} = \frac{1}{N_s} J\mathbf{v}; \qquad\qquad \Delta\mathbf{w} = \mathbf{w} - \langle \mathbf{w} \rangle; \qquad\qquad \tilde{\mathbf{v}} = J^\dagger \Delta\mathbf{w}, \qquad (75)$$

where the first and the last step are implemented with `jax.vjp` and `jax.jvp`, respectively.

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
