# Peer review of "NetKet 3: Machine Learning Toolbox for Many-Body Quantum Systems"

_SciPost Physics Codebases, doi:SciPost Phys. Codebases 7-r3.4 (2022) , SciPost Phys. Codebases 7 (2022)_

## Round 1 · Referee Report · Anonymous (Referee 1) · 2022-2-9

Strengths

1)This manuscript describes the Version 3 of the NETKET library to simulate quantum systems using artificial neural networks. The main innovation compared to version 2 is the use of the JAX python framework, which allows easy utilization of just in time compilation, of GPUs, and it provides automatic differentiation.

2) The library allows users to simulate ground states, unitary dynamics, and open system dynamics. Several primitive functionalities are exposed to the user, so that scientists can implement user-defined algorithms to tackle quantum many body problems via machine learning tools.

3) Several examples of the computational tasks are presented with sufficient details to allow new users get started with the utilization of NETKET 3.

Weaknesses

1) As the authors mention in the conclusions, there is no native support for fermionic systems.

2) As the authors discuss, one of the main strengths of NETKET 3 is that it allows users to implement arbitrary neural network structures. What is missing, perhaps, is some discussion or some guidance on which structure is more efficient or more promising for different physical models and different computational task.

Report

The version 3 of the NETKET library is described. It is implemented to be flexible, allowing users to address various computational tasks in quantum many-body physics. It provides users useful ready-to-use classes for relevant neural networks states, and also primitive tools to build user-defined algorithms.
The manuscript is well structured. It presents the design philosophy, the user interface, as well as the building blocks that can be used in a flexible manner. Several case studies are presented in a pedagogical manner.
The main innovation compared to NETKET 2 is the use of the JAX library, providing easy utilization of GPUs.
In this respect, NETKET 3 is analogous to the jVMC library (arXiv:2108.03409, Ref.[38]). However, NETKET 3 provides further functionalities not included in jVMC. A performance comparison is presented, showing a computational advantage for NETKET 3.

The use of machine learning tools to simulate equilibrium and dynamical properties of quantum systems is a growing trend in condensed matter physics. I think that the novel developments in NETKET 3 will help computational physicists to develop novel and more efficient algorithms. In this manuscript, the library is well described with various use cases. Therefore, I think that the manuscript is suitable for publication in SciPost Physics Codebases.
In the requested changes, I report a few comments that should be addressed before publication. They are aimed at improving the clarity of some parts of the manuscript.

Requested changes

1) The authors discuss the implementation of complex-valued wave-functions, mentioning the need for complex weights and biases. In Ref.[38], three main approaches (Single holomorphic network, Single non-holomorphic network, Two real (non-holomorphic) networks) to implement complex valued wave functions are described. I encourage the authors to clearly discuss the different possibilities for this task available in NETKET. A small paragraph or section about this topic would help the readers.

2) In the note 13, the authors briefly mention the possible instabilities in the calculation of the QGT. I encourage the authors to provide a more exhaustive discussion about this problem, with some guidelines to avoid it.

3) In subsection 6.1, the authors discuss a continuous-space problem. Periodic boundary conditions are implemented with a specific flag (pbc=True). It is not clear if pbc=Flase corresponds to hard-wall type boundaries. I suggest to include a specific comment.

4) The use case for continuous-variable problems corresponds to a non interacting system. I suspect that including interactions leads to more severe challenges. Can the authors provide a more challenging testbed, or some comments and guidelines on how to deal with interactions in continuous-variable problems?

5) In figure 5, it is not clear what time unit is used for the color scale of the right.

6) As the authors discuss, one of the main strengths of NETKET 3 is the possibility to implement arbitrary neural network structures. However, the authors do not provide a guideline on which structure is more suitable, depending on the problem being addressed. In the presented use cases, they sometimes employ restricted Boltzmann machines, sometimes GCNNs. Different network architectures have already been explored in the literature, e.g., dense layers, sparse networks, convolutional, graph NN (see, e.g, Phys. Rev. X 8, 011006 (2018), Nat. Comm. 9, 5322 (2018), Phys. Rev. E 101, 063308 (2020), arXiv:2011.12453, 2020), finding different performances for different problems. I encourage the authors to provide some comments to motivate and guide users in the exploration of different architectures.

7) In the conclusions, the authors might provide more comments on how they plan to implement support for fermionic systems. Notice that different approaches have been explored in the literature: Nature Chemistry 12.10, 891-897 (2020), arXiv:2112.03491, Physical Review Research 2, 10.1103 (2020), Nature Communications 11, 2368 (2020).

  • validity: high
  • significance: high
  • originality: good
  • clarity: high
  • formatting: excellent
  • grammar: excellent

Author:  Filippo Vicentini  on 2022-05-27  [id 2531]

(in reply to Report 1 on 2022-02-09)
Category:
answer to question

We thank the referee for the constructive remarks he raised, which allowed us to improve the manuscript.

Remark 1:

The authors discuss the implementation of complex-valued wave-functions, mentioning the need for complex weights and biases.
In Ref.[38], three main approaches (Single holomorphic network, Single non-holomorphic network, Two real (non-holomorphic) networks) to implement complex valued wave functions are described.
I encourage the authors to clearly discuss the different possibilities for this task available in NETKET. A small paragraph or section about this topic would help the readers.

Answer 1:

We thank the Referee for highlighting this important point. NetKet leaves the freedom to the users to use any of the mentioned architectures and strives to avoid getting in the way of users. We have considerably expanded section 3.2 to more clearly describe how networks are parametrized in NetKet and to explicitly mention the fact that any architecture can be used.

Remark 2:

In the note 13, the authors briefly mention the possible instabilities in the calculation of the QGT.
I encourage the authors to provide a more exhaustive discussion about this problem, with some guidelines to avoid it.

Answer 2:

We thank the referee for raising this issue. In fact, the instability is not associated with the calculation of the QGT itself but rather with the solution of the associated linear system of equations that would require the inversion of the QGT. To further clarify what to do when such instabilities are encountered we have shortened the footnote #13 (now #15) and added a through discussion of the tools that NetKet provides to avoid them in the paragraph titled \textbf{Linear System}.

Remark 3:

In subsection 6.1, the authors discuss a continuous-space problem.
Periodic boundary conditions are implemented with a specific flag (pbc=True).
It is not clear if `pbc=False` corresponds to hard-wall type boundaries.
I suggest to include a specific comment.

Answer 3:

As Hilbert spaces merely define the computational basis, pbc=False does not enforce any boundary conditions, so that the simulation region is effectively infinite in that case. Boundary conditions would have to be enforced at the level of a custom variational ansatz or through some operators.

We have added a comment to further clarify this point.

Remark 4:

The use case for continuous-variable problems corresponds to a non interacting system. 
I suspect that including interactions leads to more severe challenges.
Can the authors provide a more challenging test-bed, or some comments and guidelines on how to deal with interactions in continuous-variable problems?

Answer 4:

We have added an example of an interacting continuous system as Section 6.5 of the revised manuscript.

Remark 5:

In figure 5, it is not clear what time unit is used for the color scale of the right.

Answer 5:

We thank the referee for raising this issue. We updated the figure to include the unit (seconds).

Remark 6:

As the authors discuss, one of the main strengths of NETKET 3 is the possibility to implement arbitrary neural network structures.
However, the authors do not provide a guideline on which structure is more suitable, depending on the problem being addressed.
In the presented use cases, they sometimes employ restricted Boltzmann machines, sometimes GCNNs.
Different network architectures have already been explored in the literature, e.g., dense layers, sparse networks, convolutional, graph NN (see, e.g, Phys. Rev. X 8, 011006 (2018), Nat. Comm. 9, 5322 (2018), Phys. Rev. E 101, 063308 (2020), arXiv:2011.12453, 2020), finding different performances for different problems.
I encourage the authors to provide some comments to motivate and guide users in the exploration of different architectures.

Answer 6:

We thank the referee for highlighting the important question of the appropriate choice of network architecture and agree that guidelines for making this choice given a specific physical scenario would be an valuable addition to the literature. However, we believe that formulating precise and correct guidelines in a general fashion (and substantianting them with appropriate evidence) requires a considerable amount of work beyond the present manuscript and is more suited to a separate review article.

However we expanded Sec 3.2.3 (Pre-defined ansatze included with NetKet) to include an overview of the various architectures included within NetKet and briefly discussed some cases where those architectures might be beneficial.

Remark 7:

In the conclusions, the authors might provide more comments on how they plan to implement support for fermionic systems.
Notice that different approaches have been explored in the literature:
Nature Chemistry 12.10, 891-897 (2020), arXiv:2112.03491, Physical Review Research 2, 10.1103 (2020), Nature Communications 11, 2368 (2020).

Answer 7:

Since our initial submission, support for fermionic systems with a bounded number of orbitals has been added and is now part of the current stable release of NetKet (version >= 3.4).

This new functionality is further demonstrated in an added example (section 7.4) which shows how to perform a ground-state optimization for the Fermi-Hubbard model on a square lattice.

---

## Round 1 · Referee Report · Anonymous (Referee 2) · 2022-2-28

Strengths

1) Innovative topic

2) Timely toolbox for quantum many-body physics

3) Clear language and demonstrative example codes

Report

This manuscript introduces the new version 3 of the NetKet python package. The NetKet package (or library) is used to study mangy-body quantum systems by leveraging recent machine learning techniques.

Machine learning and quantum computational technologies are getting a great deal of attention because of their efficiency in solving important problems in condensed matter physics and quantum chemistry and material sciences. Therefore, this manuscript is very timely.

Moreover, it is well-written and has a broad range of introductory examples helping readers to get started with the NetKet 3 package right away (the GitHub page and the Documentation page for the package are also extremely well-constructed and easy to use).

Given the interesting topic and the clarity of the material, I recommend publication in SciPost Physics Codebases.

As a further comment, the introduction of JAX as acceleration framework and the multiple dispatch through PLUM are very welcomed additions that make this new version of NetKet much faster than before, and therefore of better applicability to large and challenging physical problems.

Minor comments and clarifications are included below.

Requested changes

1) NetKet 3 is based on JAX, a differentiable programming framework. Even though the high-level primitives as well as most of the examples shown do not require any knowledge of JAX, to really use the full potential of NetKet 3 users need to be fluent in JAX. I think this may not be stressed enough, but I may be wrong. May I suggest to add a clear sentence to Sec. 1.1 or 1.2 discussing how important a good knowledge of JAX is for using NetKet 3? (not just for building NQS, but also for the creating and debugging algorithms)

2) Linear operators introduces in section 2.2 are some of the most important objects in the library. However, I do not find in the text any explicit mention that an operator can only be defined on a fixed Hilbert space (e.g. the first argument in the constructor of nk.operator.spin.sigmax on the example of page 8). May I suggest to explicitly add a sentence on the dependence of operators on Hilbert spaces in section 2.2?

3) In equation (7) the upper limit of the sum over j, M, is not defined. It would help to have a definition of M for this simple case of a one-layer fully-connected ansatz.

4) On page 20, the first code snippet introduces F, while the second code snippet introduces g . May I suggest to add an explanation to clarify their meaning?

5) Section 4 introduces the 3 main algorithms (drivers) of NetKet 3. However, while 4.2 and 4.3 include "full" examples, 4.1 does not have one (and 4.3 actually defers the example to Sec. 8). Since 4.1 and 4.2 are rather similar in their "syntax" and the examples are not too long or complicated, would it be possible to add a full exampled in section 4.1 as well? (the one in Sec 7 is longer and more demonstrative, so I am only asking to get a short overview where all the "pieces" are shown: Hilbert, Operator, Sampler, NQS, VMC)

6) Section 4.2 comments: * add a reference to the Lindblad master equation perhaps? * typo in code snippet line 2 "latticfe" * after equation (27) the operator.LocalLiovillian is missing a "u" in the name and should be referred to also as "lind" which issued for "lind.hilbert" in the subsequent lines.

7) Section 5.1. I have a question here. NetKet has support for some discrete (only?) symmetry groups. Can this support be extended and how difficult it is? I think it would be helpful to have a sentence/paragraph related to this.

8) Continuous Hilbert spaces are a nice addition to NetKet 3. The example in section 6.4 for the harmonic oscillator modeled with a Gaussian is interesting to showcase the NetKet computational elements. I wonder if it could be more challenging to add interactions between the the oscillators so that the ground state is not just n _x = n_y = n_z =0 for all particles. Some extra comments on this section: * L=(jnp.inf..) How is an infinite length for the space interpreted, together with the boundary conditions? * Code snippet line 19: "sab" should be the sampler "sa" * In Fig. 1 the energy for the variational state starts from a really large value. What is the initialization of the Gaussian NQS?

9) In section 7.2, a full example of the optimization of a GCNN variational state for a lattice model is presented. Here I see for the first time (correct me if I am wrong) the use of "chunk_size" in the code snippet at line 21. "Chunking" is mentioned in the abstract and then never again. As far as I understand from NetKet's documentation, chunking allows to use large NN models that would not fit in memory. Since chunking is mentioned in the abstract, I would like to see a description of it in a dedicated subsection perhaps. If I have missed it, please forgive me.

10) Question about footnote 18 on page 33. Although this "non-standard MC scheme" can be used to draw (almost) independent samples, a phase of burn in is required. The code snippet on line 20 seems to imply that this burn-in phase is reduced to 0. That would not give rise to the correct sampling at the beginning. Am I missing something? Maybe related to how n_discard_per_chain=0 actually works?

11) The addition of dissipative dynamics to NetKet is very important. In fact, references 26, 26 and 27 have been published on PRL with a Veiwpoint (see https://physics.aps.org/articles/v12/74 ). From the Viewpoint I see that the last paper, the PRB one, is missing from the citations of this manuscript. Can it be added? Or is it not relevant to the techniques implemented in NetKet? Moreover, given the importance and novelty of dissipative dynamics solved with NQS, is it possible to add a figure with results in section 8.2?

12) In the benchmark section 9.2, the next to last paragraphs refers to Fig. 5, but I think it should be Fig. 4. Also, why are there two blue lines in the right panel of fig 5? In the caption, the discussion about samples should refer to the left and center panels and not the left and right panels.

  • validity: high
  • significance: high
  • originality: high
  • clarity: high
  • formatting: excellent
  • grammar: excellent

Author:  Filippo Vicentini  on 2022-05-27  [id 2530]

(in reply to Report 2 on 2022-02-28)

We thank the referee for the constructive remarks he raised, which allowed us to improve the manuscript. Below we report detailed answers to all the issues raised by the referee

Remark 1:

NetKet 3 is based on JAX, a differentiable programming framework.
Even though the high-level primitives as well as most of the examples shown do not require any knowledge of JAX, to really use the full potential of NetKet 3 users need to be fluent in JAX.
I think this may not be stressed enough, but I may be wrong.
May I suggest to add a clear sentence to Sec. 1.1 or 1.2 discussing how important a good knowledge of JAX is for using NetKet 3?
(not just for building NQS, but also for the creating and debugging algorithms).

Answer 1:

We agree with the referee that an explicit discussion of required knowledge to use or extend NetKet is helpful. It is important to note that in order to use network architectures, lattices, operators, and algorithms included with NetKet, no detailed knowledge of JAX is necessary (just like the Python interface of NetKet 2 could be used for many purposes without users having to study or modify the underlying C++ code). It is also true that in order to extend NetKet, a working knowledge of JAX is necessary. We believe it is best to refer readers to the official JAX documentation for this purpose and also provide examples on how to extend NetKet in our own documentation.

Based on the referee's feedback, we have included several sentences along these lines in sec. 1.2 to make this clear to readers early on.

Remark 2:

Linear operators introduced in section 2.2 are some of the most important objects in the library. 
However, I do not find in the text any explicit mention that an operator can only be defined on a fixed Hilbert space (e.g. the first argument in the constructor of nk.operator.spin.sigmax on the example of page 8). 
May I suggest to explicitly add a sentence on the dependence of operators on Hilbert spaces in section 2.2?

Answer 2:

We agree and have added a sentence to section 2.2 as proposed and another sentence to section 2.2.1, where we now explicitly point out the need to pass a Hilbert space object to each operator.

Remark 3:

In equation (7) the upper limit of the sum over $j$, $M$, is not defined. It would help to have a definition of M for this simple case of a one-layer fully-connected ansatz.

Answer 3:

We have added the definition of M (which is the number of hidden units) and the shapes of the weight matrix and bias vector to the main text.

Remark 4:

On page 20, the first code snippet introduces $F$, while the second code snippet introduces g . May I suggest to add an explanation to clarify their meaning?

Answer 4:

We thank the referee for spotting this. Indeed the code listing should have used $f$ as well, and we corrected the typo. We also added a sentence explaining what $f$ is after the code listing.

Remark 5:

Section 4 introduces the 3 main algorithms (drivers) of NetKet 3. However, while 4.2 and 4.3 include "full" examples, 4.1 does not have one (and 4.3 actually defers the example to Sec. 8). 
Since 4.1 and 4.2 are rather similar in their "syntax" and the examples are not too long or complicated, would it be possible to add a full exampled in section 4.1 as well? 
(the one in Sec 7 is longer and more demonstrative, so I am only asking to get a short overview where all the "pieces" are shown: Hilbert, Operator, Sampler, NQS, VMC)

Answer 5:

We thank the referee for the feedback, and have addressed the issue by adding a full example to section 4.1.

Remark 6:

Section 4.2 comments:
  - add a reference to the Lindblad master equation perhaps?
  - typo in code snippet line 2 "latticfe"
  - after equation (27) the operator.LocalLiovillian is missing a "u" in the name and should be referred to also as "lind" which issued for "lind.hilbert" in the subsequent lines.

Answer 6:

We have added a short paragraph on the Lindblad master equation as well as reference 65 and corrected the highlighted typos.

Remark 7:

Section 5.1. I have a question here. NetKet has support for some discrete (only?) symmetry groups. Can this support be extended and how difficult it is? I think it would be helpful to have a sentence/paragraph related to this.

Answer 7:

Currently NetKet only supports discrete symmetry groups isomorphic to permutations of the computational basis. Supporting symmetry groups beyond permutations would require a completely different approach and is an open research problem. To further clarify this, we have added 2 sentences stressing this limitation to the first 2 paragraphs of section 5.

Defining arbitrary permutation groups is simple, and the user must simply write a matrix whose rows define how the indices are permuted. We believe that the current manuscript already gives an example of this syntax at the beginning of section 5.1.

Remark 8:

Continuous Hilbert spaces are a nice addition to NetKet 3.
The example in section 6.4 for the harmonic oscillator modeled with a Gaussian is interesting to showcase the NetKet computational elements.
I wonder if it could be more challenging to add interactions between the the oscillators so that the ground state is not just nx=ny=nz =0 for all particles.

Answer 8 :

We added an example of bosonic particles in one spatial dimension interacting via a Lennard-Jones like potential in a periodic box, right after the harmonic oscillator example (Section 6.5). We want to point out here that this example is not considerably different to the harmonic oscillator example in the sense that most of the complications arising from particle interactions in continuous space, are taken care of within the variational Ansatz.

Remark 8b:

Some extra comments on this section:
  - `L=(jnp.inf..) How is an infinite length for the space interpreted, together with the boundary conditions?
  - Code snippet line 19: "sab" should be the sampler "sa" 
  - In Fig. 1 the energy for the variational state starts from a really large value. What is the initialization of the Gaussian NQS?

Answer 8b:

  • Infinite length is correctly interpreted as a a space with no boundaries along that dimension. Finite and infinite dimensions can be freely combined. An error is thrown if you try to set pbc=True along an infinite direction.
  • We thank the referee for spotting this typo, and we have addressed it.
  • The covariance matrix $\Sigma = TT^\dagger$ is initialized as a product of a matrix $T$ and its transpose such that it is symmetric. The entries of the matrix $T$ are initialized using a normal distribution of mean zero and variance one. A comment was added in the harmonic oscillator example specifying the exact form of the covariance matrix as well as the initialization used for it.

Remark 9:

In section 7.2, a full example of the optimization of a GCNN variational state for a lattice model is presented.
Here I see for the first time (correct me if I am wrong) the use of `chunk_size` in the code snippet at line 21. 
_Chunking_ is mentioned in the abstract and then never again.
As far as I understand from NetKet's documentation, chunking allows to use large NN models that would not fit in memory.
Since chunking is mentioned in the abstract, I would like to see a description of it in a dedicated subsection perhaps. If I have missed it, please forgive me.

Answer 9:

We thank the referee for raising the issue. We added a new section (3.3.1) titled "Reducing memory usage with chunking" discussing chunk_size and when it is useful in practice. We also added a paragraph in section 7.2 discussing why it is necessary in this case.

Remark 10:

Question about footnote 18 on page 33.
Although this _non-standard MC scheme_ can be used to draw (almost) independent samples, a phase of burn in is required.
The code snippet on line 20 seems to imply that this burn-in phase is reduced to 0. That would not give rise to the correct sampling at the beginning. 
Am I missing something? Maybe related to how `n_discard_per_chain=0` actually works?

Answer 10:

There is nothing particular about how n_discard_per_chain=0 works, and indeed without a burn-in phase the samples in the first few VMC steps might be biassed with respect to the desired distribution. However, the GCNN (and neural network states in general) are initialised with weights drawn from the truncated-normal distribution generating approximately uniform wavefunction. As the chains are initialised with samples drawn uniformly from the set of configurations, they won't be too far off from the correct distributions.

Remark 11:

The addition of dissipative dynamics to NetKet is very important.
In fact, references 26, 26 and 27 have been published on PRL with a Veiwpoint (see https://physics.aps.org/articles/v12/74 ).
From the Viewpoint I see that the last paper, the PRB one, is missing from the citations of this manuscript.
Can it be added? Or is it not relevant to the techniques implemented in NetKet?
Moreover, given the importance and novelty of dissipative dynamics solved with NQS, is it possible to add a figure with results in section 8.2?

Answer 11:

The PRB mentioned by the referee discussed supervised learning of quantum states which is not supported by NetKet at the moment. Nevertheless, we added the missing reference to the manuscript.

Remark 12:

In the benchmark section 9.2, the next to last paragraphs refers to Fig. 5, but I think it should be Fig. 4.
Also, why are there two blue lines in the right panel of fig 5?
In the caption, the discussion about samples should refer to the left and center panels and not the left and right panels.

Answer 12:

In the old manuscript section 9.1 (now 10.1) discusses Fig.4 (now figure 6) and section 9.2 discussed the weak and strong scaling of Fig. 5 (now Fig.7). As such, we don't believe that the reference to the figures were mixed.

Regarding Fig. 5 (now Fig. 7), the rightmost panel is a scatter-plot of the wall-clock time taken by all the weak-scaling simulations. As such, there are 4 points for every of the 3 implementations, but they cannot be distinguished except for the blue lines corresponding to QGTJacobianPyTree.

The motivation of the figure was to show that even if all three implementations scale relatively good undeer MPI, they have a large constant-time difference among them. To further clarify this, we have reworked the caption.

---

## Round 2 · Referee Report · Anonymous (Referee 1) · 2022-6-4

Strengths

1) Addr

Report

The revised version of the manuscript discusses interacting problems in continuous space as well as fermionic models with finite number of orbitals. These capabilities further extend the type of problems that can be addressed via NetKet, making it an extremely useful computational tool for condensed matter physicists and quantum chemists.

---

## Round 2 · Referee Report · Anonymous (Referee 2) · 2022-6-5

Report

The authors addressed all my comments. They improved the quality and readability of the manuscript with additional stylistic changes which are very welcomed.

I recommend the manuscript for publication with no additional revisions.

---

## Round 2 · Author Response

Dear Editor and referees, We are very grateful for the many comments and remarks that helped us increase the quality of the final manuscript. We have implemented corrections for all the issues you raised, and expanded the manuscript to incldue new features that were added in the last few months. Below we list the major changes to the manuscript. Of course there are many other small changes that are too numerous to list, such as updating code examples to use new or refined syntax or reflecting.

Detailed answers to all issues raised by the referees have been submitted separately as replies.

List of Major Changes

  • We have added a prominent panel highlighting the fact that advanced features in NetKet require knowledge of Jax, and linking to their documentation;
  • We have added Sec. 3.2.2 discussing the difference between holomorphic and non-holomorphic complex ansatze, as well as how their parameters as stored in PyTrees;
  • We have added Sec. 3.2.3 briefly discussing the use-case for all models included in NetKet;
  • We have added Sec. 3.3.1 discussing how to reduce memory pressure in large computations through chunking;
  • We have added a new, more elaborate example on interacting continuous systems in Sec. 6.5;
  • We have added Sec. 8 discussing how to model systems with a discrete number of fermionic orbitals;
  • We have updated the paper to refer to the current version of NetKet, 3.5, in several places. In particular, the syntax to specify the data-type of models throughout the manuscript was changed from dtype=... to param_dtype=... in order to be consistent with the same change performed by an upstream dependency (flax). The old syntax still works but will be eventually removed an year from now;
  • Various minor changes in formulations and formatting to improve the clarity of the manuscript.

---

## Editorial Decision

published